# Fingerprinting conflict: A comparative model with applications to archaeological and historical data

Søren Wichmann[1]*, Anna K. Loy[1,2], Anna-Theres Andersen[1], Ralf Bleile[3], Darja Jonjić[1,4], Jutta Kneisel[2], Norbert Nübler[4], Andrea Santamaria[1,2], Jens Schneeweiß[1,5], Gerald Schwedler[6], Katharina Zerzeropulos[1,2], Lorenz Kienle[7], Oliver Nakoinz[2,8]

1 ROOTS Cluster of Excellence, Kiel University, Kiel, Germany, 2 Institute of Prehistory, Kiel University, Kiel, Germany, 3 Museum für Archäologie in der Stiftung Schleswig-Holsteinische Landesmuseen Schloss Gottorf, Schleswig, Germany, 4 Department of Slavonic Studies, Kiel University, Kiel, Germany, 5 Leibniz-Zentrum für Archäologie / LEIZA-ZBSA Zentrum für Baltische und Skandinavische Archäologie, Schleswig, Germany, 6 History Seminar Kiel, Kiel University, Kiel, Germany, 7 Department of Materials Science, Kiel University, Kiel, Germany, 8 Johanna Mestorf Akademie, Kiel University, Kiel, Germany

* wichmannsoeren@gmail.com

**Data Availability Statement:** All data files and R scripts are available from the https://github.com/Sokiwi/ConflictPaper.

## Abstract

This paper is envisioned as a primarily methodological contribution towards a more sophisticated and systematic approach to conflict research in archaeology and history. Studies of conflicts in these fields have often focused on violence and war. Instead, we offer a more holistic approach to conflict research, taking into account different levels of both escalation and de-escalation that embrace all the possible aspects of a conflict from a mere undeveloped potential over complete annihilation to various countermeasures and stages of resolution. A model taking into account different levels of escalation and de-escalation is presented which embodies our multi-faceted view of conflicts and which also allows for a systematic, comparative analysis of conflict situations anywhere and any time in (pre)history. Through ten relatively detailed European case studies spanning the Bronze Age to the 20th century we demonstrate the comparative potential of our model and suggest ways in which it may help to identify typical patterns in conflict situations.

## Introduction

The motivation behind this paper is the realization, which has emerged over years of collaboration within our group on conflict research, that we are in need of new approaches to describing and comparing processes of conflict escalation and de-escalation, i.e. more multi-faceted ways of analyzing conflicts. Towards this end we introduce a model that treats de-escalation on an equal footing with escalation, fundamentally expanding the traditional views on conflicts. The model is applied to ten case studies from Europe spanning the Bronze Age to the 20th century, which are subsequently analyzed comparatively using quantitative methods.

**Funding:** This work was supported by the Deutsche Forschungsgemeinschaft (German Research Foundation) under Germany's Excellence Strategy (grant EXC 2150 390870439). The funders had no role in study design, data collection and analysis, decision to publish, or preparation of the manuscript.

**Competing interests:** The authors have declared that no competing interests exist.

## Theoretical foundation of conflict research

Although conflict archaeology has emerged as a sub-discipline in its own right only during the last three decades [1, 2], conflicts have a long history as one of the foci of archaeology. A wide spectrum of studies is represented in the literature, ranging from ancient or Celtic warfare [3] to 'ethnic' conflicts as in Kossinna's work [4]. However, the focus is almost always on violence and war, e.g., [5–10], with [11, 12] representing some exceptions. In recent decades, the discipline of battlefield archaeology has emerged, particularly influenced by the World Wars, initially focusing primarily on the archaeology of modern times. An emerging trend involves the investigation of prehistoric battlefields, such as the Bronze Age site at Tollense Valley [13], reflecting an expansion of the temporal scope within battlefield archaeology. The topic of war in prehistory and early history is closely related to battlefield archaeology, but by no means identical to it [14].

The concepts of 'war' and 'peace' represent significant theoretical challenges, as they have been understood and distinguished in many different ways during the history of research [15–18]. Moreover, these concepts may be conceptualized in different ways across cultures. For instance, the closest translational equivalent of 'peace' among the Ancient Egyptians is *hetep*, a concept which did not necessarily exclude violence, but rather referred to action in accord with the proper order of life on Earth and in the cosmos [19], and a frequent conceptualization of war is a hunting simile [20].

Returning to Western conceptualizations, in current conflict research there is a great variety in definitions of war, for instance including functional, qualitative, and quantitative definitions, but at least there is a tendency for states as agents to recur in some definitions [21]. Disagreement grows when anthropologists and archaeologists attempt to come up with definitions that would be valid and meaningful across cultures and time periods and agreement across disciplines seems out of reach [14] (for the discussion of this topic within archaeology, see further [22, 23]). The definitions of 'peace' can ultimately be divided into two broad categories: 'negative peace', i.e. peace as the absence of war or even violence, and 'positive peace', the processual confrontation with and active resolution of conflicts [24]. Discussions of these topics are thus often dichotomous in nature.

Instead of operating with the concepts of 'war' and 'peace' we would prefer to use the concepts of 'conflict escalation' and 'conflict de-escalation' [25], mainly in order to avoid a reduction of the focus to particular stages within a range of possible situations. Moreover, we see violence, which is a central concern of conflict archaeologists, as only one possible aspect of conflicts. Numerous other possible components are pertinent to the characterization of conflicts, including unrealized conflict potentials, threats, mediation, de-escalation, and others. The starting point for conflict research should therefore not be violence, but rather the initial stage of conflict, which is the *potential* for conflict. Some common conflict potentials are diverging needs, differing opinions, real or perceived disparities in resources, different ideologies or cosmologies or even mere misunderstandings. They are omnipresent, and they may stimulate actions either in the direction of cooperation or conflict escalation. In the course of the conflict process, new conflict potentials can arise and conflicts therefore often constitute chains of processes relating to individual conflict potentials.

Where we do specify violence terminologically as a component of our model we reserve the term for physical violence. Nevertheless, there are various non-physical forms of violence which also represent escalative behavior. As first defined by Galtung [26], *structural violence* manifests itself when a social structure or institution inflicts harm upon individuals by obstructing their ability to fulfil fundamental needs, and *symbolic violence*, as described by

Bourdieu [27], covers forms of repression that operate at a communicative level and are so subtle that they may not even be recognized by their victims.

Conflicts may involve escalatory actions that do not constitute physical violence, such as structural violence, symbolic violence or threats. However, they may also include de-escalative actions that are intended to prevent further escalation or to reduce the level of escalation. It is very important to stress that conflict, in our use of the term, is not equal to escalation. Rather, conflict potentially encompasses both escalation practices and corresponding de-escalation practices. Both tiers represent options for action. In a given conflict situation, the conflict partners find themselves at a certain level of escalation. Both have the option of deciding in favor of actions of escalation or de-escalation, although the decisions of one partner may well have an impact on that of the other.

We define conflicts as situations that give rise to specific behaviors, including escalation, de-escalation, and mediation. A conflict situation exists when at least two parties (groups or individuals) have potentially incompatible wishes or interests. Often an actor will carry out a cost-benefit analysis, assessing the threshold up to which an escalation of the conflict is justifiable and before which the conflict must successfully have been ended from the perspective of the conflict partner doing the analysis. Crossing this threshold, even success is no longer attractive because the losses are too great. Below the threshold, there is still hope that a positive balance may be obtained through a 'victory'. Such analyses, however, may be influenced by subjective or inaccurate judgements, difficulties in knowing and assessing all parameters, as well as factors not directly related to the conflict at hand, such as prestige. Honor-bound societies, in particular, seem to often cross this threshold [28]. As an alternative to the more precise cost-benefit analysis or the pursuit of 'victory' at all costs, a partner in a conflict situation may adopt the premise that conflicts in general tend to produce too high total costs or that the risk of being defeated is too probable. The strategy then is not to ruthlessly assert one's own interests, but to aim for a compromise that is acceptable to both sides. At the heart of such a strategy is the idea of cooperation, which creates a defense against individual actions having negative consequences for both partners. Targeted de-escalation may be needed to enable cooperation or at least to avoid further escalation.

In practice, these different types of action strategies are often interwoven in conflict situations. There may also be other parties not directly involved in the conflict but still potentially indirectly affected by it. Such a party may act as a mediator, perhaps in the shape of a conciliatory institution. Societies tend to defuse or de-escalate the omnipresent potential for conflict by creating institutions that can be called upon in case of conflict. These can be clan councils, councils of elders, monarchs, modern court system, etc. If the arbitration authority is generally accepted within the society, the destructive potential of conflicts is considerably reduced and the continuation of the society becomes more viable [29].

## The model

This paper mainly involves the disciplines of archaeology and history. But these far from exhaust the disciplines for which conflict research is relevant. In order to facilitate coherence in conflict research across disciplines, a generally applicable methodology which embraces both qualitative and quantitative observations, is needed. For historical conflict research, which we define as any research that examines conflicts at different times, we introduce the Escalation–De-escalation Pyramid, henceforth the E–D Pyramid, which is a model of conflicts first described by [25], cf. Fig 1. We emphasize, again, that the term 'conflict' is not to be equated with any kind of (physical) dispute between parties. In our understanding, the concept of conflict has to be situated at a more abstract level and encompasses any situation where

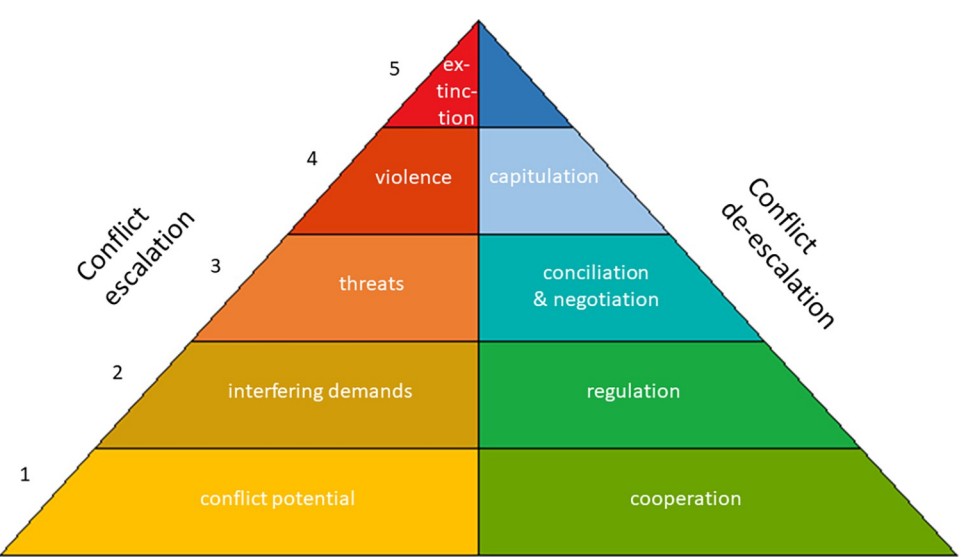

**Fig 1. The Escalation–De-escalation Pyramid (E–D Pyramid).**

different, perhaps competing, interests of at least two conflict partners are present. Hence, conflicts start out with a conflict potential. How this conflict potential is dealt with and unfolds itself depends on a variety of factors. Escalation is only one possible strategy and can take place across several stages. Importantly, escalation has to be considered together with its opposite: de-escalation. We assume that an instance of escalation can be turned into an instance of de-escalation at almost any point in time if appropriate measures are taken. Furthermore, an entire conflict can be resolved at any stage of escalation or de-escalation, although this is difficult since it usually involves finding a balance of multiple interests. The extinction of a conflict party represents the most extreme and only irreversible conflict resolution. Our view of conflicts is embodied in the E–D Pyramid.

The E–D Pyramid is perhaps best explained in more detail through an example. For this purpose we offer a deliberately simplified thought experiment. Let us assume that there are two prospective conflict partners on an island. Partner A and B might both like to have the island for themselves. This brings us to the first step on then conflict escalation side of the E–D Pyramid: *conflict potential*. The island with its resources cannot belong entirely to either A or B at the same time. Hence, A and B have two very basic options, both of which require action: They can escalate the conflict by stating their demands in a mutually exclusive way ("This island belongs to me and me alone!"), leading to the second level of the E–D Pyramid on the conflict escalation side: *interfering demands*. Then again, one of the conflict partners might withdraw from the claim to the island, or they might agree to cooperate on the exploitation of the island's resources, switching over to the conflict de-escalation side: *cooperation*. On the second level of the E–D Pyramid the de-escalation strategy is called *regulation*. In our example, this might mean establishing some rules ("This part of the island is yours and this part is mine"). Further escalating the conflict, however, leads to *threats* on the third level of the E–D Pyramid. On this escalative level one or both conflict partners try to use threats of, for instance, physical or financial harm to make the opposition concede ("I will hurt you if you continue to exploit my island!"). Structural violence [26] could be understood to reside on this level. On the de-escalation side of the same level, conflict partners might start using the strategy of *conciliation and negotiation*. This strategy may profit from an external input like a mediator or arbitrator. Following Glasl ([30], based on [31]), such a role is even mandatory. Successful

negotiations or a successful conciliation process lead to the establishment of mutually agreed upon rules ("You get to exploit these resources, however I get to exploit these other ones"). Should the conflict be escalated further, the fourth level on the escalation side of the E–D Pyramid, called *violence*, is reached. Here at least one of the conflict partners is actively harming the other. To de-escalate, the equivalent strategy would mean *capitulation* (flight, annexation, e.g.). The ultimate escalation would be the *extinction* of at least one of the conflict partners (partner A killing partner B—if need be in a self-destructive manoeuvre). At this level there is no de-escalative counterpart since one (or perhaps even both) of the partners has (have) gone extinct. Generally speaking, the costs for the conflict partners rise with each level reached on the E–D Pyramid.

As far as we know, conflict processes do not have to follow certain trajectories on the E–D Pyramid but can jump steps and change between escalative and de-escalative actions at any time. It is, however, an interesting empirical question whether there are some patterns that are more frequent than others.

Stepping up or down on the E–D Pyramid may be accompanied by psychological effects controlled and exploited within a power dynamic. A radicalization of sufficiently many members of the group(s) involved in the conflict will facilitate escalative processes, maybe even allowing for the skipping of one or more steps on the pyramid (e.g., p. 417 in [32]). Correspondingly, processes increasing understanding, and sometimes the development of a sense of community, help in facilitating de-escalation.

In spite of the simple thought experiment given above, which involved just two conflict partners, there is nothing that hinders an application of the model to more complex scenarios involving multiple partners. Interpretations then also become more complicated and perhaps more prone to ambiguities. For instance, given a conflict between A and B, should we interpret the event of an alliance between A and a third partner, C, as representing escalation or de-escalation? The decision would depend on whether the event is more likely to aggravate the situation or not, which is of course a matter of interpretation. The model also allows for flexibility with respect to the point of view. In the case studies of this paper we try to take a neutral point of view, looking at conflicts from the 'outside'. But a conflict might also be modelled from the 'inside', taking the view of a particular partner. Let us again refer to the fictional example with the newly formed alliance of A and C. Even in the case where this is expected to bring more stability and is therefore deemed de-escalatory from the point of view of the entire situation, it may still be seen as escalatory from the point of view of C in case this partner was not in any way affected by the conflict prior to the alliance.

As researchers, we usually do not directly observe the presence of a certain level of the pyramid in some historical situation. Instead, we observe *indicators* of a given level. The indicator concept functions as a Middle Range Theory that links the observations with a 'high-level theory', the latter being the conflict theory. It is up to our interpretation to assign a given indicator to a particular level. Often assignment can be ambiguous, as illustrated by the following three examples. Physical evidence for a political border in the archaeological record may signify 'interfering demands', corresponding to Level 2 on the escalation side of the pyramid, but it may also imply 'regulation', corresponding to Level 2 on the de-escalation side. An execution site may imply escalation in the sense of the exercise of violence, but also de-escalation, inasmuch as it may have arisen as part of an effort to protect the population through a code of rules [11]. Processes cementing a group identity may de-escalate potential quarrels within the group, but may also escalate group behavior towards other groups. Much of this paper will be devoted to the identification and discussion of such *indicators* (for more on this concept see next section).

While it is fair to assume that for a given conflict there is some degree of interrelatedness of actions corresponding to the various indicators observed, i.e. a network of causality—perhaps with some events that are more central than others—it not a requirement of the model that two indicators are in fact related. For instance, in a given conflict some act of threat (escalation Level 3) may be identified as well as some act of violence (Level 4), and both would then be registered in the descriptive model. But the description is a static one, so there is no necessity of an assumption of direct causation between the act of threat and the act of violence. In a further development of the model a dynamic perspective may conceivably be introduced, allowing for the identification of diachronic patterns of conflict developments. This is a matter for future work, however.

In order to put the comparative potential of the model to the test we offer a range of case studies from diverse disciplines that are analyzed from the perspective of the model. The aim of our model is to facilitate the comparison of conflict situations across historical periods, areas, and datasets/disciplines. We are aware that the further back in time a case study extends, the less precise the analysis of conflict dynamics is expected to be. Still, we deem it worthwhile to take up the challenge of comparing situations as far removed as, for instance, Bronze Age Crete and 20th century Russia, even if we know less about the former than about the latter. In terms of time spans, our case studies represent anything from several decades to several centuries. We believe that similar dynamics are expected to play out in short-term conflicts and longer-term ones, which is why we decided not to introduce distinctions in the temporal dimension. But we are aware that more research would be needed to fully justify this decision. Most cases exclusively concern whole groups (societies), but our model is impervious to whether agents are constituted by individuals or groups. That said, we regard history as driven more by groups than by individuals.

If, as we hope, it turns out that the different cases can meaningfully be compared at the level of abstraction assumed by the model, we will have achieved our main aim with this work. We would furthermore hope to have paved the way towards generalizations regarding expected conflict trajectories, although we realize that solidly establishing such generalizations is not within reach of the present study.

## Methods

Having introduced our general model we now go on to describe how it is applied when studying particular cases and when subsequently comparing the cases. As researchers we make empirical observations, typically of artifacts and written sources when it comes to disciplines such as archaeology and history. We then interpret these observations as manifestations of various actions. In the context of conflict research we understand actions as normally representing a conscious intervention implying some strategy, concept, attitude, etc. Using the pyramid model, these interventions can be seen as belonging to different levels of the escalation or de-escalation side of the pyramid (perhaps more than one simultaneously), i.e. to different cells in the model. We refer to a physical manifestation of a particular pyramid cell as an *indicator*.

What counts as an indicator for a particular cell in the model is to a large extent up to the individual researcher to decide, but the general keywords in Fig 1 provide some guidelines. In Table 1 we moreover provide a few examples of the kinds of indicators that may be found in the different cells, although we would like to stress that they should not be seen as normative.

While we cannot attempt to comment in detail on all the indicators listed in Table 1, there are a couple which seem particularly worthy of added remarks. The appearance of 'high population density' at escalation Level 1 and 'population pressure' at Level 2 are not intended to mean that increasing population density necessarily cause escalation, but it has been suggested

**Table 1. Examples of indicators at different levels of the E–D Pyramid.**

|  | Escalation | De-escalation |
|---|---|---|
| Level 5 | **Extinction**<br>• Demolition of settlements | **[Perhaps never realized]**<br>• Suicide? |
| Level 4 | **Violence**<br>• Injuries<br>• Damage to weapons<br>• Vandalism of statues<br>• Abrupt changes of use for sites | **Capitulation**<br>• New use of buildings<br>• New social structure<br>• Tribute system<br>• Assimilation |
| Level 3 | **Threats**<br>• Offensive weaponry<br>• Offensive fortifications | **Conciliation and negotiation**<br>• Settlements protection by clustering<br>• Settlements protection by scattering<br>• Hidden settlements<br>• Defensive fortifications |
| Level 2 | **Interfering demands**<br>• Population pressure<br>• Perceived inequality | **Regulation**<br>• Physical borders<br>• Standardized coinage<br>• Symbolic fortifications<br>• Public sites for negotiation processes<br>• Ritual sites for regulation<br>• Division of labour<br>• Monuments symbolizing institutions |
| Level 1 | **Conflict potential**<br>• High population density<br>• Locally restricted resources<br>• Ideological differences<br>• Inequality | **Cooperation**<br>• Trade<br>• Coinage<br>• Identity-establishing monuments |

that the probability of disputes may rise with larger population sizes [33, 34], and population pressure is known to sometimes give rise to territorial disputes [35]. Next, 'demolition of settlements' (escalation Level 5) cannot automatically be regarded as a result of conflicts. For instance, a temple might have been ritually destroyed after having fallen out of use or a building might have been demolished simply because it was already in a poor condition [36]. Accordingly, a destroyed building should only be interpreted as an indicator of conflict escalation if other archaeological indicators—such as the presence of scattered weapons, skeletal remains, or traces of looting—support such an interpretation.

The advantages of the indicator concept presented here are obvious. It is simple, applicable across disciplines and makes conflicts both qualitatively and quantitatively comparable. Since very heterogeneous observations from different disciplinary domains are projected as indicators onto the same simple model of the E–D Pyramid, similar conflict scenarios can be recognized despite their different individual characteristics and the comparison of heterogeneous conflict scenarios should help to understand how the different types of conflict work.

While assignment of indicators to cells of the pyramid is categorical in nature, each assignment is accompanied by a numerical value corresponding to the reliability of an indicator as evidence for the corresponding level of escalation or de-escalation. This value is called the *confidence score*. Confidence scores may be displayed as points on a scale of hue in order to aid visual interpretation. If there is no evidence for the indicator as representing a given cell, the assigned value is 0 and the corresponding color is white. Increasingly high numbers, which run up to 1, may be displayed as increasingly dark hues. When multiple indicators are present their values must be concatenated. This can be done in different ways, as discussed below.

When assigning confidence scores to each cell in a matrix the researcher should follow the guidelines in Table 2. Although the table shows selected stations on a cline, any point on the

**Table 2. Guidelines for assigning confidence scores (CS): Five equi-distant points on a scale from 0 to 1 and their interpretation.**

| CS | Interpretation |
|---|---|
| 0 | The observed (de-)escalation actions cannot be adequately displayed by the given indicator |
| 0.25 | There is no clear relationship between indicator and cell, however it is not entirely ruled out |
| 0.5 | There is a probable relation |
| 0.75 | There is a very probable relation |
| 1 | This indicator has a really clear relationship with the pyramid cell |

scale can be chosen. For instance, a value of 0.95 would translate to a very high level of certainty still leaving a bit of room for doubt.

One indicator may potentially be interpreted as providing evidence for different levels of escalation or de-escalation or even for both escalation and de-escalation at any of the available levels (when one indicator is assigned to different slots with different confidence scores, the slots can be thought of as being joined in a fuzzy set in the sense of [37]).

As mentioned earlier, when more than one indicator is in play, there is a need for aggregating values across the cells of the E–D Pyramid in order to achieve a single fingerprint for the entire conflict situation. Different methods of aggregation highlight different aspects of the interplay between different indicators and their confidence scores. Some simple aggregation methods that suggest themselves are: sum, max, mean, and median. Each method and its consequences for interpretation can be described as follows.

*Sum*: When summing up the confidence scores, the number of indicators for the given E–D Pyramid cell has an effect. It is therefore assumed that each indicator provides added, independent evidence for a given cell in the Pyramid. A large number of indicators for a cell, even if each confidence score may be weak, will typically indicate that the existence of the level of escalation or de-escalation in question is not only well supported, but also particularly intense, i.e. has played a major role.

*Max*: The maximum value represents the single, strongest indicator for each pyramid cell. Using this value means that other conflict components are ignored.

*Mean*: The mean value uses all indicators for pyramid fields and averages them. Only empty pyramid cells and the zero values are excluded from the averaging process. This places less importance on individual indicators (when more than one is present), which is appropriate under the assumptions that the data available are uncertain or not representative of the entire situation or that there are possibly different, unconnected situations involved.

*Median*. This is similar to the mean, but more resistant to outliers.

Visualization of the fingerprints, i.e. the E–D Pyramid based on aggregated values, is a handy way of summarizing a conflict. Here different cells of the pyramid are colored with different intensity depending on the certainty of assignment. This allows different fingerprints representing different conflict situations or possibly also different aggregation methods to be quickly compared. The underlying numbers, however, are part of the fingerprints and provide the basis for further quantitative analyses. When we go on to look for patterns in the set of conflict case studies we need to refer to vectors of 1–10 numbers each that are the mathematical representations of the fingerprint. These vectors allow for various exploratory analyses.

## Data

The following pages contain ten European case studies from which the data for this paper is extracted. The case studies follow in a rough chronological order and pertain to the last four millennia, i.e., from the Bronze Age to the 20[th] century. Each case study was produced by a

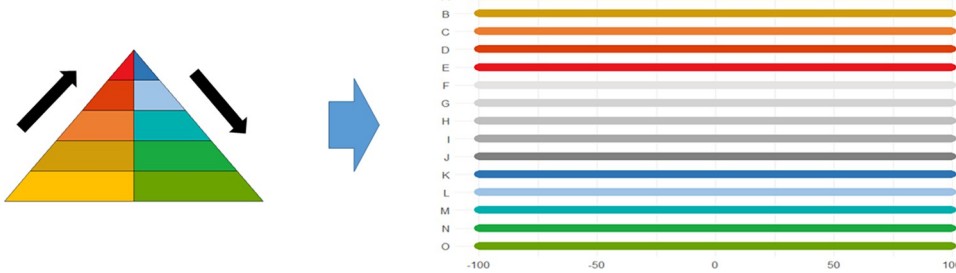

**Fig 2. Symbolization conventions from pyramid to timelines.** A-O are indicators occupying all the different cells of the pyramid.

specialist from the team of co-authors and can be seen as having been sampled randomly from the immense amount of information on the human past, although the expertise available of course ultimately guided the temporal and geographical selection, the latter of necessity being Eurocentric. While a lot of individual leeway was allowed for, all case studies contain some standard elements. *Subtitles* heading each case consist of (a) an identifier signalling the main geographical area or topic, (b) a time period (in parenthesis), and (c) 2–4 keywords. The identifiers recur in analyses of the data. In the text for each case study *some words are highlighted in bold*. These are words that make reference to indicators of escalation or de-escalation. In online data files accompanying this paper are *lists of indicators with assigned confidence scores and dates*. It is not guaranteed that all indicators in the list are mentioned explicitly in the text, but when they are mentioned, they will be signalled by the bold-faced text. This informally applied highlighting helps the reader to maintain a focus on major elements feeding into analyses. Finally, each case study contains an *illustration naming the indicators and graphically providing time lines*. The times lines are also given as dates in the online.csv data files. These files should be consulted for the full information, because the graphic timelines sometimes sacrifice details for the sake of clarity. Thus, when one indicator is involved in more than one pyramid cell, it is the color corresponding to the cell at the highest level which is chosen for displaying the timeline. When two cells, respectively pertaining to escalation and de-escalation, both represent the highest level, a gray-scale color is chosen rather than one symbolizing either escalation or de-escalation. The darkness corresponds to the level. The timelines are organized in the top-down vertical dimension according to the route corresponding to a climb up and down the pyramid starting at the escalation side, with the grayscale colors symbolizing ambiguity of escalation vs. de-escalation sandwiched in between. These conventions are illustrated in Fig 2. We display years BC as negative numbers, and events of an instantaneous nature, typically lasting less than a year, are symbolized as large dots.

Although the case studies are generally separate and self-containing, the two first overlap temporally and are also parallel in the sense that they discuss similar kinds of archaeological data; moreover, the narratives unfolded in some case studies happen to mention some of the same historical events or protagonists, such as the Teutonic Order to which one case is devoted but which is also mentioned in the case study dealing with Sambia.

## Schleswig-Holstein (Bronze Age): Cooking pits, weapons, and fortifications

Only a few Bronze Age sites provide clear evidence of the use of violence, such as the battlefield of Tollense Valley in Mecklenburg-Western Pomerania (e.g., [38]), and conciliation is even more difficult to register in an archaeological context since only the material remains can be used to identify conflicts and conflict regulation in prehistoric periods. Swords, lance heads,

and arrowheads functioning as weapons represent the threat of violence. Defensive weapons such as shields, helmets or body armor may be interpreted as a reaction to the threat of violence, but at the same time serve as protection against actual violence. Fortified settlements may be understood as a defense and also as a regulatory measure in the sense of territories. Burnt horizons with weapons and skeletons found in these settlements indicate violent destruction [39, 40]. The laying down of weapons—so-called hoards or deposits—are evidence of an end to the threat of violence emanating from the weapons and can be interpreted as pacification. Gathering places, such as large cooking stone pit areas indicating joint activities, are evidence of cooperation and certainly also serve to initiate agreements and thus regulations.

The following section describes the cultural background. In Northern Europe, the Bronze Age (1700–500 BC) began relatively late—a good 500 years later than in Central Europe. The first bronze-processing groups in Central Europe (Únětice, ca. 2200–1600 BC), who produced the Nebra Sky Disc among other objects, controlled the metal trade for a long time. The Jutland Peninsula and northern Lower Saxony were **cut off from the bronze resource**, while Mecklenburg-Western Pomerania was included in the metal exchange [41, 42]. This changed with the collapse of the Únětice occupation in the south around 1600 BC. The north experienced an upswing, which is reflected above all in **rich grave goods** and a rapid increase in bronze finds from around 1500 BC. From the Older (1700–1100 BC) to the Younger Bronze Age (1100–500 BC), a strong **social transformation** takes place, which is particularly evident in the graves. In the Older Bronze Age, large burial mounds were the dominant burial construction, in which only part of the population was buried with rich grave goods. Weapons (dagger, sword, and lance) and jewellery are common burial objects, while female graves with jewellery are less common [43, 44]. In the Younger Bronze Age, urn graves with fewer grave goods dominate. This continues to be jewellery (mostly small pieces), but also objects for personal use such as dress pins, buttons, razors, tweezers, and awls. Tools and weapons are rare. The urn graves contain all age groups and sexes and indicate a rather **egalitarian burial custom** [45, 46], which contrasts with the Older Bronze Age burial mounds interpreted as elite burials [47, 48]. The transition from the Older to the Younger Bronze Age is thus characterized by a social change, which is mainly reflected in the burial customs. Settlements in both periods, on the other hand, consisted of individual farmsteads with or one or two longhouses, which were scattered across the Jutland peninsula within a few kilometres of each other. It was not until the Iron Age (from 500 BC) that the settlement pattern changed in favor of smaller houses in hamlets and villages. Fortified settlements are missing and signs of boundaries and fencing [49] are rare. In contrast, increasing demarcation can be observed in the Iron Age from 500 BC. Houses and farms were fenced in and settlements were also demarcated from the surrounding countryside [50]. The field systems changed and towards the end of the Bronze Age, so-called Celtic Fields slowly developed, some of whose clear field boundaries are still visible in the forests today [51]. Some fortifications were built on the Jutland peninsula after 300 BC, as well as barriers known as *hulbælter* [52–54].

With this background on the settlement and burial patterns, the various indicators of conflict and de-escalation mentioned above should now be considered. The distribution curves for weapons, cooking stone pits and depositions are based on the aoristic method [55–57], which enables the statistical distribution of unequally dated finds to be visualized on a time axis (Fig 3).

Weapons as an indicator for the threat of violence, especially swords and lances, dominate in the Older Bronze Age of Schleswig-Holstein. For swords there is evidence of traces of use [59], but whether in combat or for sport/games is unclear. There are also some finds pointing to swords as having prestige character rather than serving as tools for direct combat [14, 60, 61]. The **increase in weapons** and metal finds from 1500 BC onwards can be explained on the

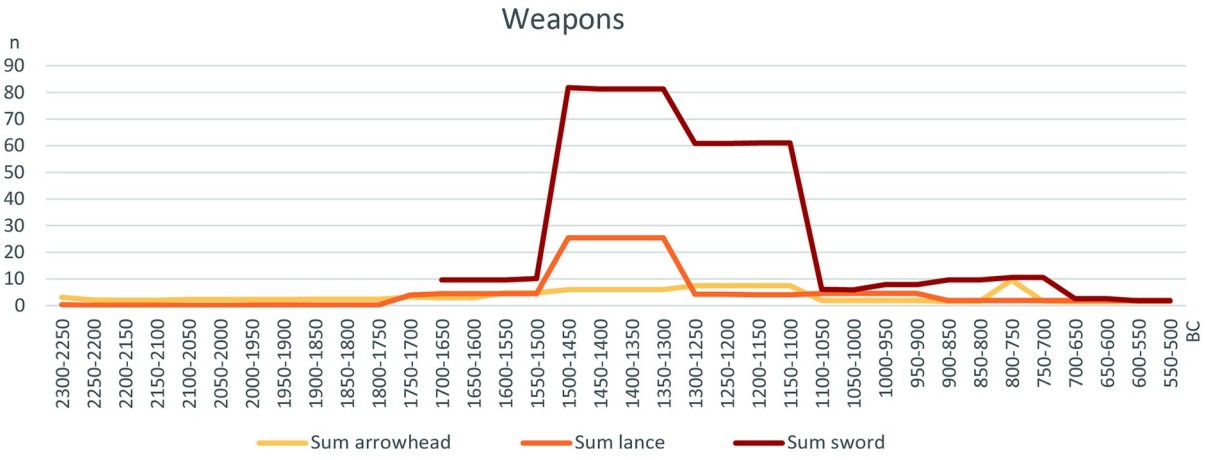

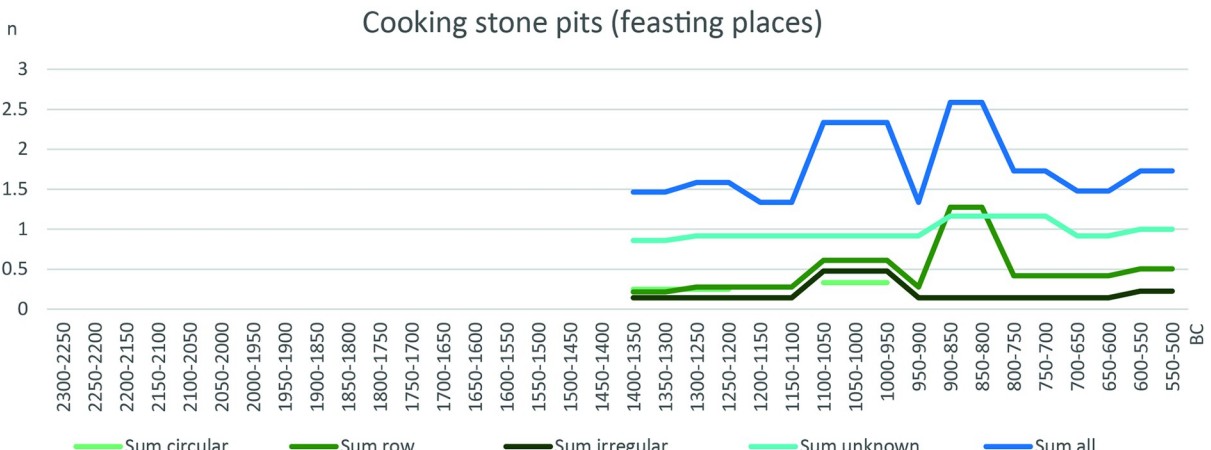

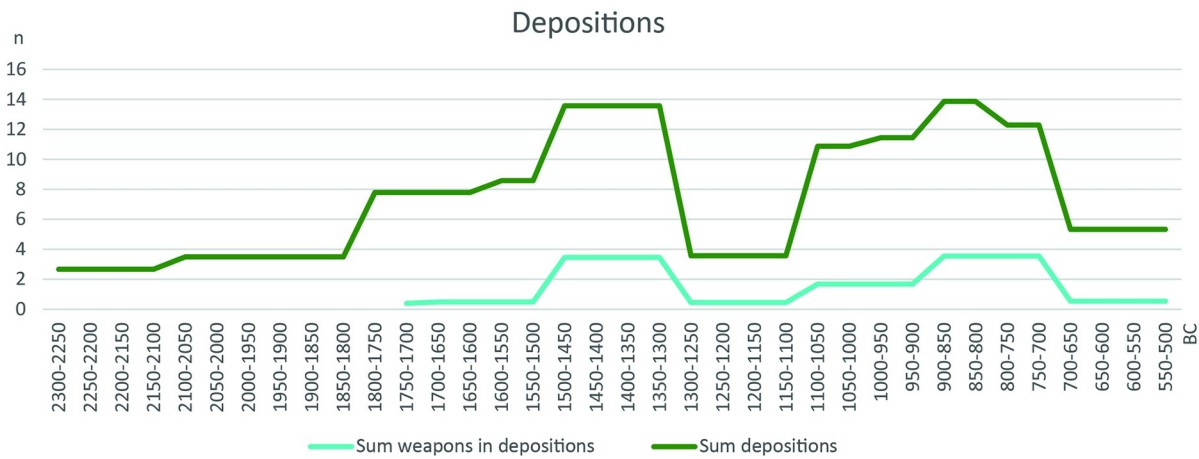

**Fig 3. Aoristic distribution of weapons (sword, lance, arrowhead), cooking stone pits and deposition finds in Schleswig-Holstein during the Bronze Age.** (After [58]).

one hand by the now **free access to metal resources** in the south [62]. On the other hand the connection to supra-regional networks also seems to have created **potential for conflict,** prestige and increasing social inequality. De-escalation measures are rare, apart from the **depositions**. The depots mainly contain jewellery and tools and, if weapons, lances rather than swords, but they were never pure weapon hoards. It can therefore be assumed that they are rarely directly related to the ending of conflicts. Nonetheless, to a lesser extent, these sometimes jointly performed depositions can also have a regulating effect. Towards the end of the Older Bronze Age, from around 1300 BC, cremation burials slowly became common although the construction of burial mounds continued and even increased slightly. More people were now buried in the elaborately constructed burial mounds and special grave goods such as gold and amber even increased slightly (Figs 3, 4 in [57]). The **social inequality** that is evident in the labour-intensive burial mounds for a small part of the population disappears with the urn graves from around 1100 BC, in which a large part, if not the entire population is now buried [46]. During this period, a new phenomenon can be observed, namely **Cooking stone pits**, places outside the settlements that were regularly visited. Their proximity to water holes/springs and the strongly heated stones, which give the phenomenon its name, indicate the preparation of food as part of festive activities (e.g. [63–65]). There is an increase in the number of sites at the end of the Older Bronze Age. This is precisely when we see the period of change in burial customs, the decline in weapons and rich grave furnishings. It can be assumed that the meeting places were used not only for festivities but also for negotiations and agreements between the widely scattered individual farmsteads.

In the Younger Bronze Age, the amount of weapons declines sharply, which is partly due to the change in burial customs (a sword does not fit in the urn). Nevertheless, a **slight increase in weapons can be observed from around 900 BC**, this time not only in graves, but also in deposits. In Schleswig-Holstein, a few **pure weapon depots** can be observed [58], some with the laying down of several swords, which indicates a deliberate laying down of weapons by several people, and thus an end to the threat posed by the weapons. The frequency of **depositions generally increases** in this period, even if most of them contain jewellery, the deposition of bronze objects apparently playing an important role in society.

Further changes can be observed around 900/800 BC. The first **defensive weapons** appear in northern Europe [58, 66–68]. On the one hand, they prove the existence of violence, but on the other hand they represent a countermeasure against physical attacks. It is also the period in which we observe the development of various **centers of wealth** in northern Europe, in which particularly rich grave goods, intensive southern contacts and resources such as amber or foreign burial customs such as house, face and box-shaped urns and the renewed construction of a few large barrows can be observed. In Schleswig-Holstein this is Albersdorf; other centers are on Fyn or in Brandenburg [69–71]. At the same time, we see a renewed increase in the number of cooking stone pit sites, indicating an intensification of communal activities between individual farmsteads. This changes fundamentally at the end of the Bronze Age from 500 BC, when the first **fences** appear [72], individual farmsteads are abandoned in favor of hamlets, **Celtic Fields** structure the landscape [50], and new large urn cemeteries are used communally.

In summary, we observe an older Bronze Age with much violence emanating from weapons, but few regulatory measures. The question therefore arises as to whether these weapons were really used for physical violence or rather had a regulating effect in the sense of competition. It is only in the later Bronze Age that we see numerous regulatory measures, ranging from feasting places to the deposition of weapons, and the appearance of defensive weapons. With the beginning of the Iron Age and the end of individual farmsteads, the regulation measures increase significantly once again, as is to be expected when larger groups live together.

A list of indicators is provided in Fig 4.

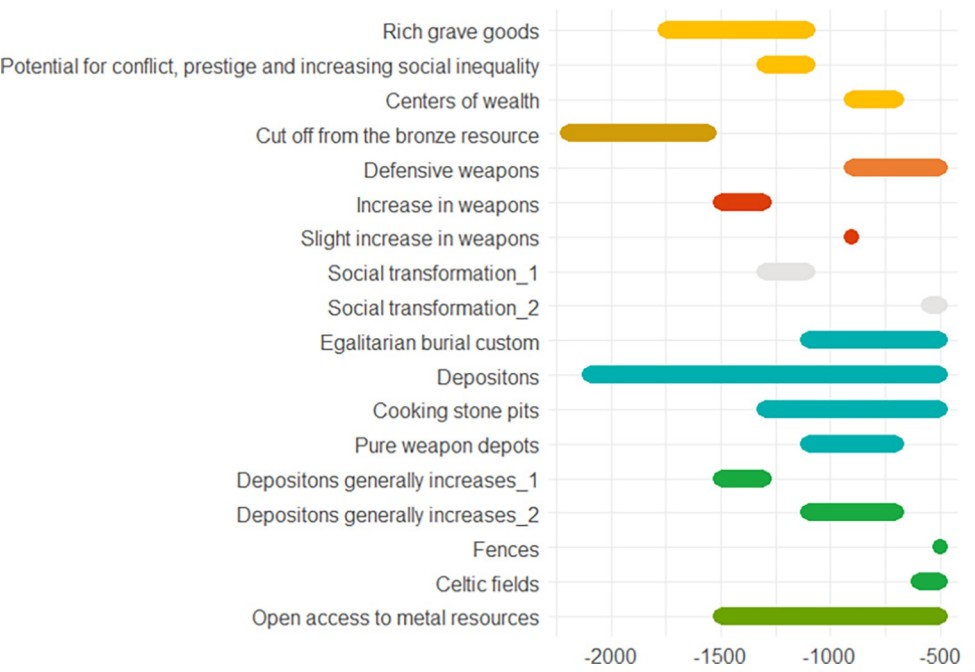

**Fig 4. Approximate chronology of the indicators described in the Schleswig-Holstein case study.**

## Poland (Bronze and Iron Age): Weapons, fortifications, and boundaries

In the Early Bronze Age, the western part of Poland belonged to the area of the early bronze-processing Únětice groups (2200–1600 BC), while the north still belonged culturally to the southern zone of the Nordic area of influence. In the east, the Trzciniec culture emerged from around 2400 BC (2400–1300 BC), which remained largely metal-poor. The southern influence of the Tumulus Culture with its small burial mounds is recognizable in the Middle Bronze Age. From around 1300 BC, the Lusatian groups emerged, which are known above all for their vessel-rich grave goods. The different regional groups existed until the Iron Age (500 BC) and extend into present-day Brandenburg and Saxony. The burial customs and settlement structures are very different in the individual areas. In contrast to Schleswig-Holstein (see previous case study), however, numerous fortified sites, some with dense inner housing structures, are known from both the Early and Late Bronze Age and Early Iron Age [73–75]. In the following, the focus lies primarily on western and northern Poland. Although the weapons and fortifications were collected for the whole of Poland, the main areas of distribution are in the regions west of the Vistula and south of the Warta River [76–81]. The only exceptions are the swords, which mainly originate from depositions in Pomerania [80]. According to the current state of research, the fortified sites are limited to the areas between the Oder and Vistula rivers and south of the Warta River. They extend further into Mecklenburg and are occasionally found at the edge of the northern Carpathian Mountains (Fig 8 in [58]). An overview of the hoard finds is still lacking, but the weapons mostly come from hoards and are only rarely found in graves (see Fig 5). Gathering places in the form of cooking stone pit areas are documented [K. Dzięgielewski personal communication, 82, 83], but overview studies on this topic are missing. There are also indications of later ritual use of the fortifications, but this has only been proven for a few sites in the research area [84, 85]. In this context, a row of palisades from Dyrotz near Potsdam, Brandenburg, is also of interest. It runs parallel along a depression and consists of a row of palisades firmly anchored in the ground, as well as a row of pits [86, 87] and lies

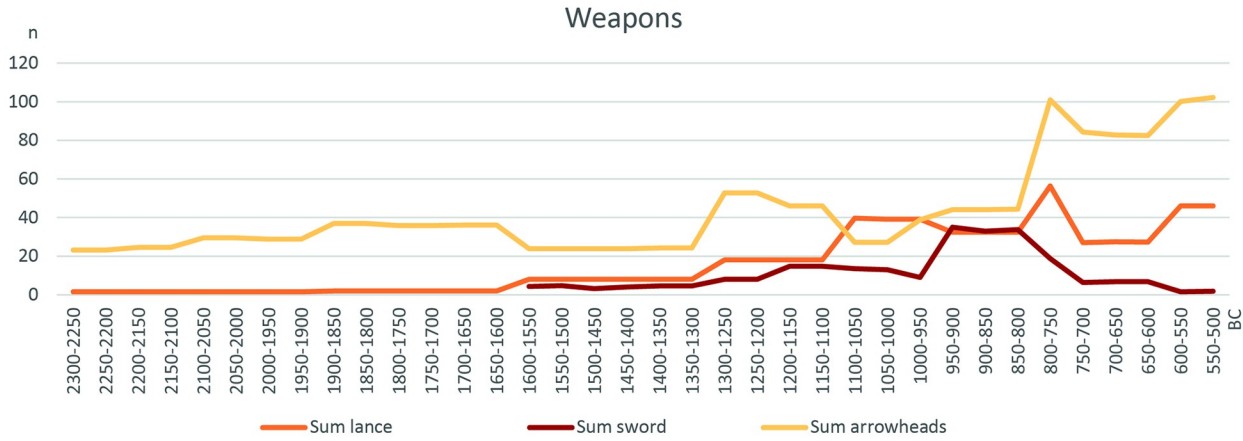

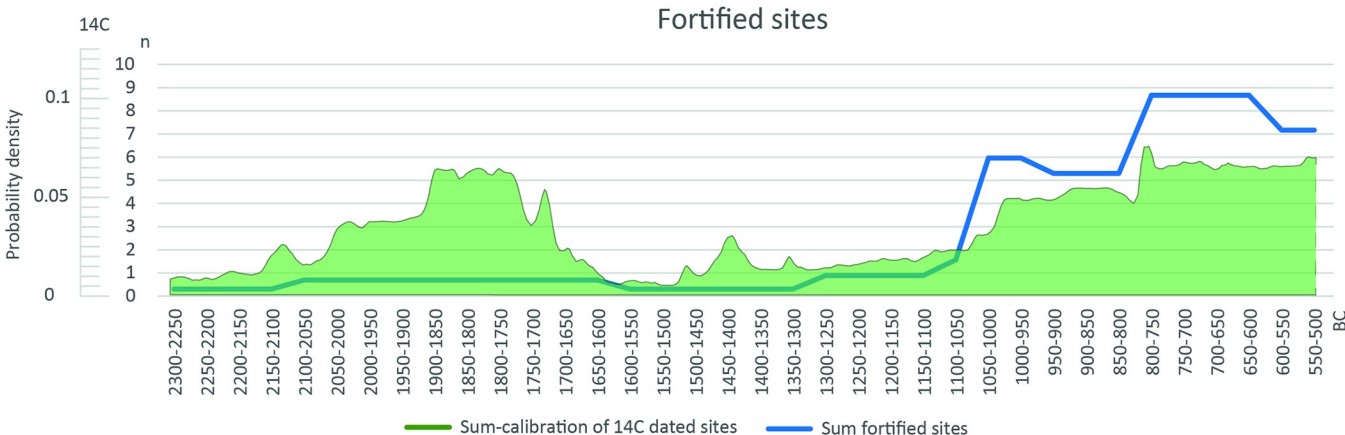

**Fig 5.** Upper panel: Aoristic distribution of weapons (sword, lance, arrowhead) and fortified settlements in Poland during the Bronze and Early Iron Age. (After [58]). Lower panel: General sum calibration of $^{14}$C-dates for fortified sites. (After [90]).

approximately on the border between the eastern Lusatian groups and the north-western area of influence of the Nordic circle. Dendrochronology places the site in the 13th century BC. It can be compared with land ditches and pit alignments that are known from Saxony and Saxony-Anhalt and, like networks, demarcate and delimit areas. Similar to the cooking stone pits, the earliest dates for the pit alignments point to the 14th century BC [88, 89].

Against this context of settlement and fortifications, the various indicators of conflict and de-escalation mentioned above will now be considered. As in the Schleswig-Holstein case study, the distribution curves are based on the aoristic method [55–57]. A sum calibration of the $^{14}$C data was also created for the fortified settlements (Fig 5).

In terms of **weapons**, there is a clear correlation between arrowheads and fortified settlements in the Early Bronze Age and lances and arrowheads at the beginning of the Early Iron Age. This recurring pattern shows that ramparts were attacked with projectiles rather than swords, which is also known from other sites in Germany [91]. An analysis of the find contexts confirms the picture, as the majority of the projectiles, unlike in Schleswig-Holstein (see previous case study), originate from settlements or **fortified settlements** (Fig 6). Although the number of swords also increased in Poland from 1100 BC, the majority of these came from depositions in Pomerania, a region in which fortified settlements are still lacking (Fig 8 in

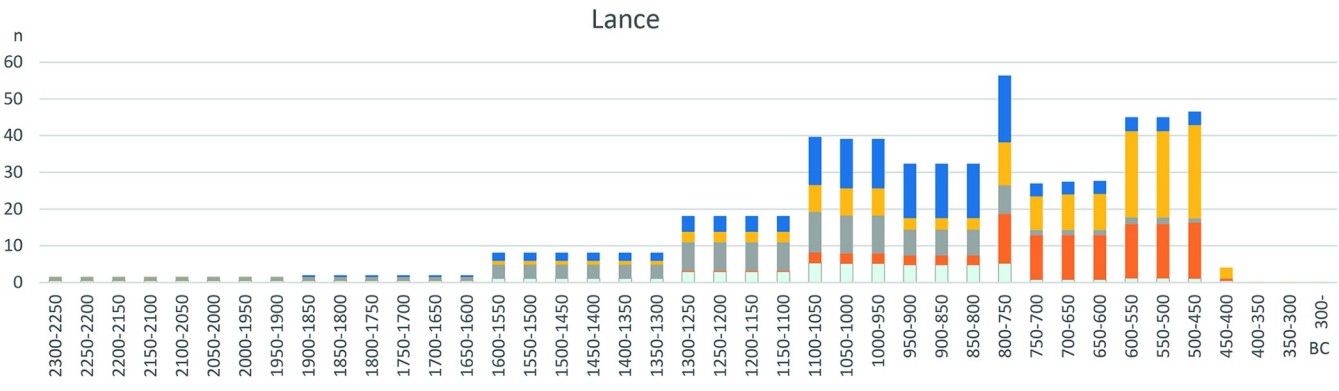

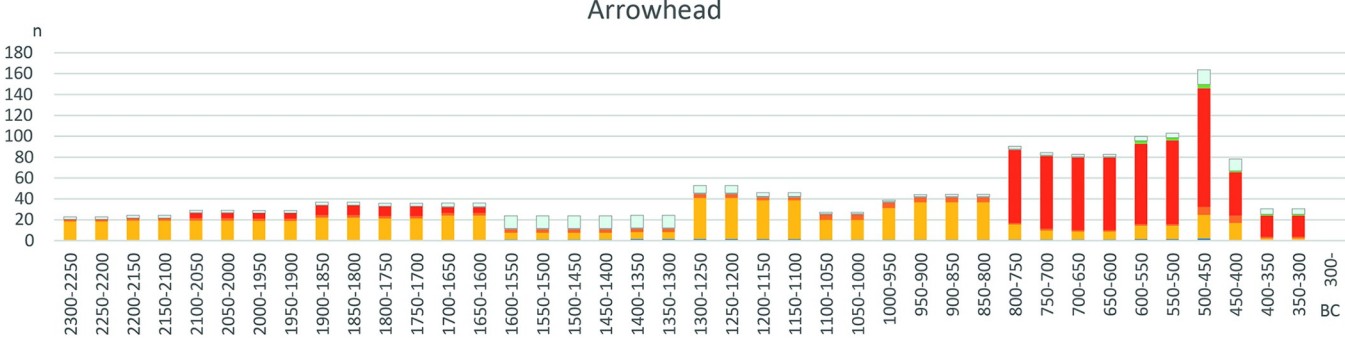

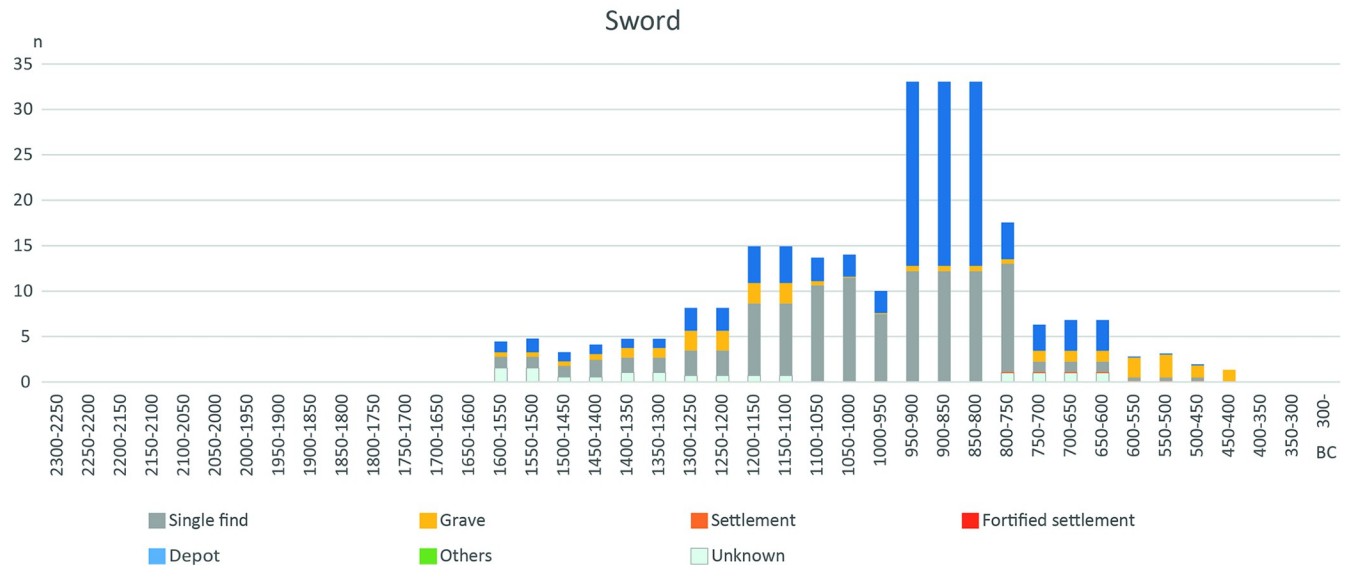

**Fig 6. Aoristic distribution of the different context of the weapon finds in Poland.**

[58]). The violence and threats emanate from the **projectiles**, and the fortifications provide protection against them. The so-called **Biskupin-type settlements** form a separate group, which were only inhabited for a very short time between the middle of the 8th century and the beginning of the 7<sup>th</sup> century BC and whose distribution extends from Brandenburg to Greater Poland and Kujawy [74, 75, 92, 93]. A large number of lances and arrowheads originate from them. Over 260 such weapons are documented from Biskupin alone [76, 77]. The distribution of arrowheads in Biskupin, for example, is clearly concentrated on the edge and gate area and indicates a possible **attack scenario** (Fig 9 in [58]). The **massive expansion of the fortifications** at the end of the Bronze Age and the beginning of the Early Iron Age demonstrates a considerable potential for conflict. The actual outbreak of violence can be documented in the **burned horizons** of the ramparts, partly with skeletal finds in Wicina or Słupca [94–97] in the Iron Age. In the Early Bronze Age settlement of Bruszczewo, too, the arrowheads tend to be located at the edge of the fortification. The most recent phase is dated by a fire in the gate area as well (Fig 9 in [97]).

Based on the weapons and settlement structures, two scenarios can be identified, one for the Early Bronze Age and one for the end of the Bronze Age to the Early Iron Age.

Only a few fortified sites have been recorded from the Early Bronze Age. They were mostly used over several centuries [98] and belong predominantly to the Únětice groups. Four sites are located on a north-south axis that extends from Greater Poland to the Moravian Gate and are relatively regularly distanced from each other. Three further sites are located on the northern edge of the Carpathian Mountains, also at regular distances from each other (Fig 8 in [58]). All sites are located on the border between the metal-rich Únětice groups in the west and the metal-poor Trzciniec culture in the east. The fortified settlements were intensively involved in exchange and trade and presumably also controlled it [99, 100]. Metal production, innovative casting techniques, and rich large burial mounds demonstrate the interconnectedness of the settlements. The example of the Bruszczewo site shows the different contacts to the west (ceramic forms, burial customs, house construction), to the north (amber as a resource) and to the south (imported ceramics, amber, metal as a resource) [99]. It can be strongly assumed that the fortified settlements, which are otherwise absent in the western Únětice area [101], form a **clear boundary here, which on the one hand controls trade, but on the other hand also represents a demarcation to the east**. The fire horizon at the end of the settlement and numerous individual human bones in the settlement layers, mostly adult males as far as can be determined, indicate a possible **violent end to the settlement** [97]. After the collapse of the Únětice culture around 1650/1600 BC, these settlements in Poland were also abandoned and evidence of a subsequent colonization of the region is sparse. It can be assumed that the end of the Únětice groups north of the middle mountain range was followed by a **hiatus** of one to two centuries [41, 102]. The first early Middle Bronze Age finds come from the south [103]; in the actual core region, a permanent settlement can only be observed again during the second half of the 15<sup>th</sup> century BC. This gap is also reflected in the weapon curve (Fig 4), which only increases again from 1300 BC. This is a period **without fortified settlements**, but characterised by **major changes**. **New exchange networks** are established [104], **the subsistence strategy changes** [105, 106], and small burial sites are abandoned in favor of large urn cemeteries (e.g. Kausche, cf. [107]). This is the period in which the first **defensive weapons**, helmets and shields, are also found in Poland [66, 108] and **weapons are increasingly found in depositions**. The **battle of Tollense Valley** in Mecklenburg-Western Pomerania [38] and **the palisade of Dyrotz** [86], both of which are not located in Poland but neighbor it to the west, fall into this period. There is obviously a renewed need for demarcation. The **construction of fortified sites resumed** after two centuries from around 1000 BC. Some of the fortifications were located in the catchment area of the Oder and probably ensured the exchange of

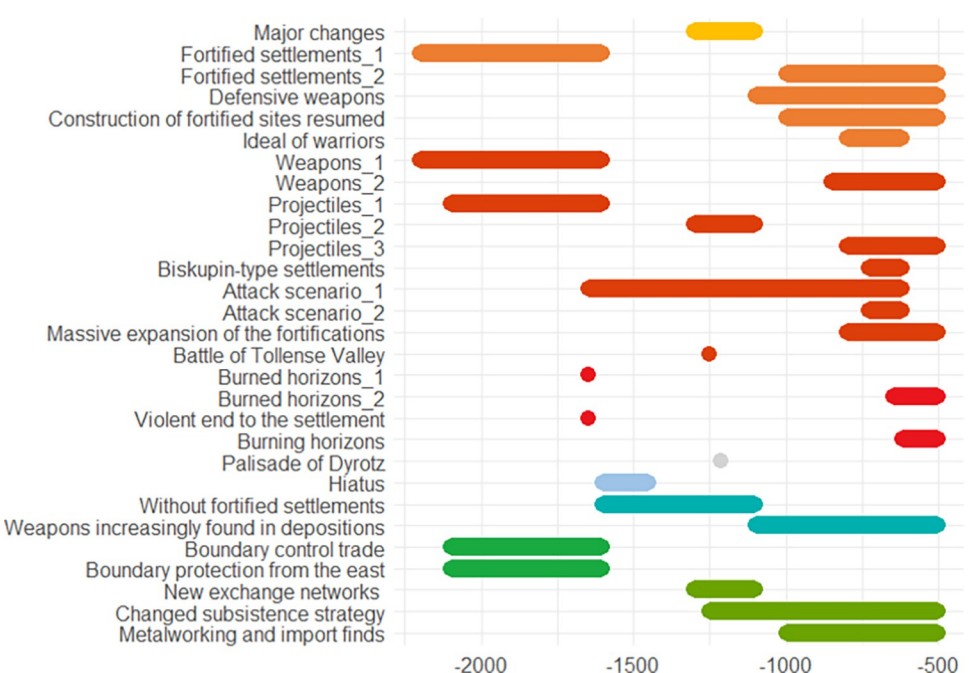

**Fig 7. Approximate chronology of the indicators described in the Poland case study.**

goods between the south and the north (Fig 8 in [58]). Evidence of **metalworking and import finds** can be regularly observed in the fortified sites [109, 110].

From around 850 BC, with the beginning of the Early Iron Age, there was another **sharp increase in the construction of fortified sites of the Biskupin type**. They only existed for a short time and are evidence of a dense and often structured concentration of a large number of people [74]. For people to live together in this type of settlement a great deal of **cooperation** is required. At the same time the community offers greater protection against possible attacks. Towards the end of the Early Iron Age and in the phases thereafter, **burning horizons** indicate the violent end of some fortifications, sometimes, as in Słupca, with **human skeletons** found in the cultural layers or pits [96]. During this period, an increase in pictorial sources can be observed, which, in addition to hunting, also depict fighting or an **ideal of warriors** [111–114].

In Poland we see similar patterns in the different time periods. Regulation and protection against violence go hand in hand with a higher potential for conflict and more weapon finds and, at the same time, larger exchange networks. In comparison to Schleswig-Holstein (see previous case study), where the potential for conflict appears to be rather low and rises at the end of the Bronze Age, a recurring sequence of conflicts and de-escalation strategies can be observed in Poland from the Early Bronze Age onwards, increasing steadily towards the end of the early Iron Age, perhaps connected with the Scythian expansion to the west. An overview of the indicators is provided in Fig 7.

## Crete (Early to Middle Bronze Age): Elite competition and the emergence of literate administrations

This case study illustrates how the appearance of writing in Europe is tied to a transition between escalating and de-escalating strategies. The earliest traces of writing in Europe occurred on Crete at the dawn of the 2nd millennium BCE. Around the 19th century BCE, two

mature systems are attested, namely Cretan Hieroglyphic and Linear A. These have normally been seen as separate writing systems, although the boundary between them are nuanced, and reciprocal influences can be inferred [115, 116]. They flourished in the frame of the emergence of sophisticated administrations in the Middle Bronze Age (MBA) Crete. Here we argue that the development of these literate administrative practices in the Middle Bronze Age (MBA) Crete were part of an active strategy of a leading elite to sustain de-escalation following conflicts among elite groups.

The beginning of the 2<sup>nd</sup> millennium BCE is considered a transitional phase between the 'Prepalatial' and the 'Protopalatial' period, the latter being archaeologically defined by the construction of central-court buildings in urban centers. This transitional phase has been seen as the peak of a period of elites in competition, struggling for access to networks involving overseas communities [117]. This led to territorial divisions within the island [118] and a general reorganization of social life [119].

Fig 8 represents an attempt to depict this complex dynamic by the diachronic manifestation of indicators. Expressed in terms of Minoan relative chronology the period in question spans the Early Minoan (EM) IIB to the conventional end of the Protopalatial at the end of the Middle Minoan (MM) II. The central time period, the EM III, and, in part, the MM IA, are characterized by the accumulation of indicators suggestive of conflict potential and competing demands. In this period notable **population growth** coincides with a decline in commercial activities, and perhaps with shifts in climate. For instance, at Knossos, the growth rate jumps from 0.125–0.250% per year in EM I-II (with a population of around 2,600) to 0.335–0.582% in EM III-MM IA (with a population ranging from 6,000 to 11,100). Similar trends are registered in other later palatial centers like Mallia and Phaistos [120]. Some scholars suggest that

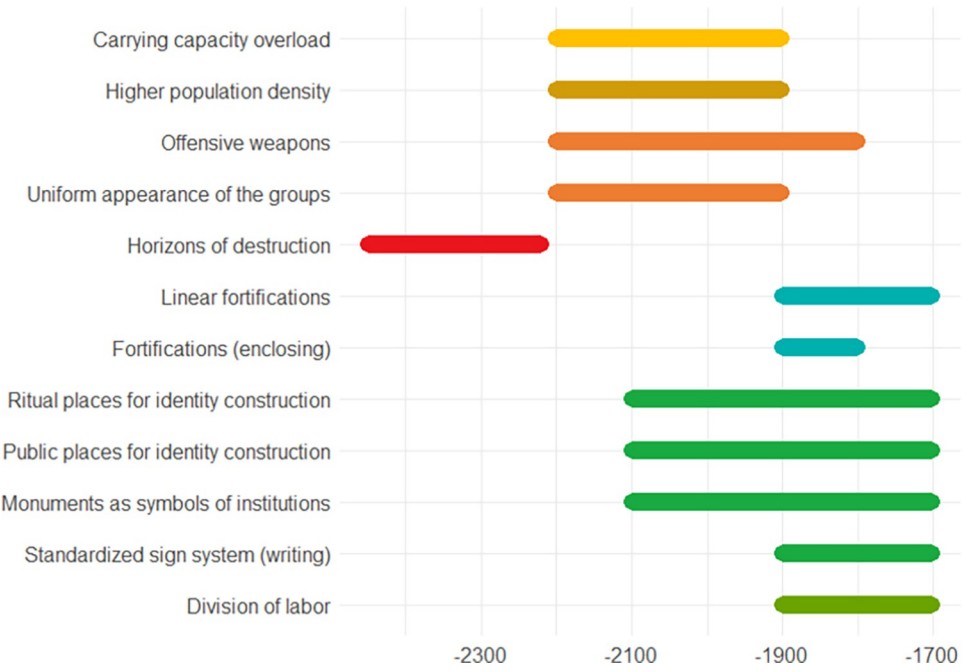

**Fig 8. Approximate chronology of the indicators described in the Crete case study.** Since precise dating is often challenging, absolute chronologies must be considered tentative. The dynamics of developments at different sites vary, for instance with a certain degree of continuity observed at Phaistos [124], in contrast to a more abrupt change, perhaps following a horizon of destruction, at Mallia [125]. Furthermore, the onset of the Protopalatial period differs from site to site, with ranges spanning several centuries from the palace's foundations dating to the MM IA period or even the EM III at Knossos and Mallia, while they date to the MM IIA at Petras.

rapid aridification over one or two generations poses a significant disruption to agriculture in late Prepalatial, likely resulting in a population exceeding the carrying capacity [121]. As Barry Molloy (in p.c. to A. Santamaria) points out, however, the resolution of these works is too low and their conclusions are contradicted by data from Western Crete. Nevertheless, it is worth noting that soil exploitation increased significantly during the MM IB [122, 123], reaching levels unmatched even in the modern era. This likely reflects the necessity of supporting a growing population.

Indicators for conflict potential are accompanied by indicators suggesting a further escalatory dynamic. Burials from this period exhibit a notable increase in the frequency of **offensive weaponry**, contrasting with the scarcity of such weapons in MM II burials [123, 126, but cf. 127]. During the late Prepalatial period, approximately 80% of the island's metallurgical activity was devoted to weapon production, a unique case in the Aegean context [128]. Around the end of the Early Bronze Age (EBA), 'long daggers' become progressively prevalent in Minoan metallurgy over 'daggers' less suited for combat or not employed as weapons at all [129]. During the MM II period, offensive weaponry is still present, but its relative frequency compared to both the preceding and succeeding MM III periods is undoubtedly lower. (The discovery of two swords at Mallia suggested a reorganization of violence that favored the acquisition of skills through longer training and the production of weapons requiring a more complex labor organization [123, 130]. It is worth noting, however, that these objects are prestige items whose gold decorations are incompatible with mass production).

Such escalating dynamics are evident in the taxonomy of seals that provide the earliest examples of writing, known as the 'Archanes formula'. Almost all the material from EM III-MM I exhibits two distinct stylistic trends, termed 'Border and Leaf Complex' and 'Parading Lion/Spiral Group', respectively. They represent two internally cohesive and **uniform social groups** [131, 132], coherent with the idea of competing elites at the end of the Prepalatial. The Archanes formula is limited to seals of the 'Border and Leaf Complex'. The stylistic uniformity of each of these divergent trends is compatible with an escalation at the third level of the pyramid.

The dawn of the MBA is widely seen as a transformative period archaeologically defined by innovative paradigms in residential organization and ritual practices. These findings have been interpreted as reflecting the emergence of new shared values and social norms for identity negotiation [133]. During this period, clear indicators of de-escalation compatible with the escalatory levels reached in the Prepalatial period are observed. Notably, there is a move towards creating **public spaces for identity negotiation**, involving larger segments of society [119]. Palaces, primarily **ceremonial structures** [124, 134], exemplify this trend, with their west courts forming "a kind of buffering area between the town and palaces while serving as gathering spaces for the urban population" (page 368 in [135]). They testify to the gathering together of communal practices by multiple corporate groups or households to foster group identity and perhaps develop hierarchy [119]. From EM III-MM IA onwards, both burial sites and urban areas witness a trend towards larger-scale practices, evident in ceramic assemblages, including tableware, and notable in burial sites like Archanes Phourni, Myrtos Pyrgos, and Platanos [136–138]. The rapid increase of 'peak sanctuaries' during the Protopalatial also underscores the communal nature of regional cult activities. Significantly, collective practices are often closely linked to the erection of palaces, as evidenced by the building of the NW terrace at Knossos [136] and the activities taken on before the construction of the Phaistos palace [139]. While the most notable signs of monumentalization in palatial contexts emerged during the MM II period, early instances can be recognized in the construction of Tholos B and Tomb 3 at Archanes (late MM IA) and the similar monumental building of Chrysolakkos (MM IA-B). They mark a process of "institutionalization of social inequality" under the

influence of Knossos and Mallia's palatial centers (page 108 in [137]). This shift is also manifested in landscape use emerging from surveys, which is characterized by an explosion of small sites (farms and field sites) in marginal regions. This is interpreted as more intensive exploitation of agricultural resources, possibly indicative of a centralization of power or other trajectories of hierarchization [119, 123, 140]. Palaces often overlap with pre-existing urban focal points, reinforcing the idea of continuity, and suggesting a desire for communal legitimacy rooted in the past. The absence of signs of destructing or discontinuity could imply that higher levels of escalation were absent or rare.

The development of palatial centers led to more complex organization and **labor division**, reflected, for example, in individual mason's marks on building blocks possibly specific to certain groups of craftsmen and the establishment of large workshops like the one in the Quartier Mu at Mallia, catering to standardized production under elite control. Notable outcomes of this standardized production are the seals of the 'Mallia Steatite Group', known for their high stylistic uniformity and widespread use in administrative contexts [141]. These seals are found throughout the island, and their iconographic and stylistic components exercised a wide influence on the contemporary Minoan glyptic. Urban planning becomes evident in the tendency toward centripetal stratification of settlements, where key buildings cluster around a central core [135]. Simultaneously, a fortification pattern emerges, which can be seen as part of a de-escalative dynamics in the framework of [25]. Especially from the MM IB period onwards, defensive structures such as enclosure walls and towers were erected, serving as deterrents and defining boundaries (e.g., [142]). Although earlier examples are sporadically attested, limited to eastern Crete, this situation clearly contrasts with that of the Prepalatial period [118]. Moreover, Prepalatial enclosure walls are unlikely to have had a defensive function [143]. These defenses are complemented by 'guard houses' following the roads' path, which serve as pivotal points during pacification and support central authorities [118]. Occasionally, these structures are associated with towers and bastions interconnected by walled walkways. Crucially, guard houses are exceptionally rare during the late Prepalatial period. This was likely due to the limited effectiveness of such structures during times of instability [144].

RevealinglyRevealinglyRevealingly, **standardized writing** on both seals and administrative documents may serve as an indicator for a de-escalating strategy. The 'Archanes formula' already points to an ongoing de-escalation around 2000 BCE. Seals with the formula are found alongside seals belonging to the 'Parading Lion/Spiral Group' in Ossuary 6 at Archanes Phourni. Intriguingly, they were still deposited in non-communicating rooms (rooms not sharing access): Room 3 (alongside two 'hybrids' in bone with shapes typical of the 'Parading Lion/Spiral Group', i.e., CMS II,1 no. 386 and 388) and Room 1, respectively. By the MM II period, writing became a prominent feature on seals, primarily used for administrative purposes in palace-related contexts, reflecting increasingly complex recording systems. Despite inter-regional variations, evidenced by the use of Cretan Hieroglyphic and Linear A, there are strong traces of exchanges, interactions, and hybridizations among different literate administrations [115]. Hieroglyphic administrations in the northeastern part of the island, corresponding to the territorial units around the palaces of Knossos, Mallia, and Petras, exhibit high consistency in the usage of writing. Indeed, they share specific writing sequences known as 'formulae'. Formulae likely indicate administrative functions [143]. They are both incised on clay administrative documents and engraved on seals according to precise criteria reflecting an organized and widely recognized hierarchy. For example (referring to signs here through standard transnumeration), only the most frequent sequence, i.e. CH 044–049 (and, in very rare cases, CH 038–010[−31]), may occur alone, while CH 044–005 may only co-occur with other formulae, indicating a function that necessarily presupposed other ones [143]. Rarer and more complex sequences that presuppose a higher status are typically found on seals made of

semi-precious hard stones, indicating a higher status. In spite of these, high-status seals do not show huge differences from a stylistic perspective, but rather a higher quality of the manufacture, which points to uniform production choices crossing social categories [116]. Features of seals belonging to the 'Border and Leaf complex' are usually continued during the Protopalatial, while those of the 'Parading Lion/Spiral Group' are largely abandoned. This situation starkly contrasts with the conflictual heterarchy emerging from the glyptic of the preceding phase, in which seal impressions are extremely rare [145]. As a result, these literate administrative practices indicates the development of standardized tools for institutionalizing a social and administrative hierarchy on a broader scale and the efforts toward a shared management of resources and economic interactions.

Overall, this dynamic shows that the advent of writing in Crete occurred as the culmination of an escalating process between two unified elite factions, evident in the contrasting uniform groups reflected by contemporary glyptic, which are mirrored by indicators of conflict potential (i.e. a higher population density and an unequal access to resources) and violence (i.e. offensive weaponry). Subsequent to the construction of palaces, the integration of writing within a literate administration likely signifies the acknowledgment of the social group responsible for its introduction, aligning with a series of de-escalative indicators pointing out that the elites went through a phase of conciliation (i.e. defensive fortifications and public places for negotiation), regulation (i.e. division of labor, monuments and linear fortifications) and cooperation (i.e. sites to establish identity).

## Sambia (c. 900 BC-1300 AD): Amber, strongholds, and trade

Sambia, or Samland as part of former East Prussia, is a peninsula measuring around $75 \times 30$ km in the Baltic Sea between the Vistula Lagoon, Curonian Lagoon and the associated spits. In the north and west, mainly cliffs border the Baltic Sea, while in the inland south and east, the Pregel and Deime rivers form the border. Since 1945, the Sambia has belonged to Russia as the Kaliningrad Peninsula (Kaliningrad Oblast'). Sambia is one of the most amber-rich coastal areas of the Baltic Sea and has played a central role in the amber trade since prehistoric times, especially in the Roman Iron Age. This is one of the origins of the so-called Amber Road. It has a special significance in many periods. The high density of fortifications in Sambia is particularly striking. Rich resources and many fortifications make this landscape predestined for a case study on the course of conflicts from an archaeological perspective.

> The [...] Estland [= Eastland] is very large, and there is **very many a stronghold** [*burh*], and in each stronghold there is **a king** [or ruler]. And there is very much honey and fishing, and the king and the most powerful men drink mare's milk and **the poor and the slaves** drink mead. There is very much [or great] **conflict** between them. And there is not any ale brewed among the Ests, but there is **mead** enough. (page 16 in [146], our emphasis).

We owe this contemporary assessment of the 9[th] or early 10[th] century to Wulfstan the navigator, apparently an Anglo-Saxon with Scandinavian experience [147], who was travelling along the southern Baltic coast from Hedeby eastwards to Truso. His travel accounts, as well as those of another trader, Ohthere of Hålogaland, were included in the *Old English Orosius*. Wulfstan's mention of the constant conflict between local rulers has shaped our view and has long been associated with the numerous fortifications, whose origin was thus assumed at the transition to the early medieval Prussian culture of the Viking Age and the subsequent so-called *spätheidnische Zeit* (Late Pagan Period) of the 11[th] to 13[th] centuries [148]. One reason for Wulfstan's journey could have been amber, even if it is not mentioned directly and Wulfstan's profession must remain open in the end.

The natural amber deposits of the Sambian peninsula were of **great economic importance** in all archaeological periods. **Large-scale amber trade** began in the Bronze Age, as evidenced by the numerous finds of Baltic amber outside its natural habitat. Examples include the Aunjetitz (Únětice) culture in Bohemia and Mycenae in the eastern Mediterranean [149]. Initially, Jutland played a leading role as the region of origin, but from the Late Bronze Age at the latest, Sambia took a dominant position in the amber trade. From the Early Iron Age onwards, the amber routes moved from west to east into the Baltic region. Trade connections can be traced throughout Europe [150, 151]. The early importance of Sambia is reflected in the **exceptionally high number of bronze finds** in the Early Bronze Age (Fig 50 in [152]). It can only be supposed that this is an early manifestation of the amber trade, since raw materials for local bronze production are absent in the area. The largest concentration of Bronze Age cemeteries and hoards can be found near the coast in western Sambia, where the largest concentrations of fortified sites are located as well. Open settlements are underrepresented. A few deposits from Period V of the Bronze Age have been found, mostly deposits from bronze casters (e.g. the casters' deposit at Littausdorf, Fischhausen district, with numerous semi-finished weapons and jewellery (Fig 63d in [153]). The **number of hoard deposits** increases significantly up to the Early Iron Age, which is generally seen as an indication of turbulent times. Until the Roman Iron Age, cremation burials predominated in Sambia; it was not until the Roman Iron Age that inhumation burials can be found, before cremation burials became prevalent again in the Migration Period.

The 1st-3rd century, the Early Roman Iron Age, is considered the 'golden age' of Sambia, at this time inhabited by the so-called Dollkeim-Kovrovo-Culture [154]. **Amber trade** with the Roman Empire led to a period of **prosperity and wealth**. It is even assumed that the special character of Sambia was only due to the **Roman imports** and the orientation towards Roman standards [154]. Therefore, Sambia has already been described as a port of trade with the Roman Empire [155]. Amber trade allowed for rich imports of non-ferrous metals, which were a prerequisite for **rich jewellery**, which is particularly characteristic of the West Baltic tribes. For the Romans, amber was a luxury item. Pliny Secundus (the Elder) writes in Naturalis Historia XXXVII: "Among **luxury objects**, it [amber] is valued so highly that even a small image of a human being is worth more than the price of living healthy people." (page 402 in [156]). In Sambia, on the other hand, amber does not seem to have been highly valued, although it was the source of wealth. At any rate, this is how Tacitus assesses it in Germania, chapter 45, from the Roman perspective; he describes the *Aestiorum gentes* as **peaceful amber collectors**: "The Aestians themselves do not use it [amber] at all; they pick it up, raw as it is, bring it unprocessed to the merchant and marvel at the price they are paid for it" (page 403 in [156]). In fact, amber does not appear to have played a significant role in the Sambian Iron Age Dollkeim-Kovrovo culture. On the other hand, there are many **Roman imports**, especially Roman mosaic beads, millefiori beads, and beads with gold foil, but **Roman weapons** also form an important group of finds. Their frequency decreases after the 3rd century and amber beads and pieces of raw amber are increasingly found instead.

In the 4th century, the Late Roman Iron Age, the importance of Sambia as a cultural and economic center in the Eastern Baltic temporarily declined. In the course of the Marcomannic Wars at the turn of the 3rd and 4th centuries, the Sambian-Roman **connections broke off** almost completely. Roman imports are missing from the Sambian burial grounds, and finds of amber in the Roman trading center Aquileia also decline. In addition to changes in the course of the Amber Road [154], the **migration** of the Goths to the south-east is considered responsible for this, with finds of Baltic jewellery in burials of the Chernyakhov culture in the North Pontic area forming the basis for the assumption that people from the Baltic moved with the Goths [157]. During the Migration Period, Sambia was once again the cultural and economic

center of the Eastern Baltic [158, 159]. The burials from the Early Migration Period are once again **richly equipped**. The **amber trade** is revived with a **new exchange network**, the trading partners now being the Germanic tribes in the west [154]. Despite this clear break between the Roman Iron Age and the Migration Period, it is assumed that there was **a continuity of settlement and population** at least from the Roman Iron Age until the conquest by the Teutonic Order in the Middle Ages. It is assumed that the Prussians, who were first mentioned in the 9th century, developed directly from the Dollkeim-Kovrovo culture [154].

Contacts with Scandinavia intensified during the Viking Age. As early as the 9th century, Scandinavians and Prussians co-existed in Sambia. A long settlement continuity is assumed for the local Prussians, which is supported above all by the **continuous occupation of the burial grounds**, sometimes from Roman Iron Age until the Late Middle Ages, while the Scandinavians dominated the trade. The role of amber in Viking Age trade can only be inferred indirectly. Numerous finds of amber products and half-fabricated items from the Anglo-Scandinavian period from York (9th-11th century) make an import from the Baltic region likely (page 31 in [147] with further references), so that we can probably assume that **amber was traded** during this period. In Sambia, **amber is found in fortifications and settlements** (e.g., [152, 160]), but also in non-agrarian settlements, such as Kaup-Wiskiauten, probably the most important center of Sambia at that time. In this respect, the numerous fortifications can possibly be linked to the amber, because participation in and control over the amber trade probably implied the existence of **wealth** that had to be protected [148]. Incidentally, we learn about the end of Kaup from the written sources: They report on **war raids** by the Norwegian king Haakon Haraldson against Sambia around 950, which prove that there was definitely a threat, and reasons for precautions. However, the relationship between the Prussians and the Scandinavians seems to have been **largely unproblematic** in the Late Pagan Period. The **conquest** of the area by the Teutonic Order in the 13th century was less peaceful. The knights took power by force when they introduced Christianity. It is known from written sources that the Prussians repeatedly fought back with **uprisings** [152].

Sambia belongs to the West Baltic cultural area. A significant number of **weapons have been found in Sambia in all periods since the Bronze Age**. The vast majority come from burials, but hoard finds also play a role. The weaponry initially consisted of a battle axe, lance and sword, later developing into the socket axe (a.k.a. *Tüllenbeil* or *Tüllenaxt*) and short sword. Swords and axes declined, while lances remained in use throughout the period. In the pre-Roman Iron Age, there is evidence of **shields and lances**. The Roman Iron Age is characterized by rich equestrian burials. The burials of men contain mainly lances and rectangular shields as well as spurs and sometimes socket axes. **Swords are very rare** in the graves of the Dollkeim-Kovrovo culture [161]. Simple, relatively small dagger knives first appear in the Late Roman Iron Age, which then developed into stabbing weapons with a longer reach. **Dagger knives** are often regarded as the main weapon of the Balts in the Migration Period. The most common combination of weapons in men's graves is the combination of combat knives, or later **dagger knives, with spears and lances**. Shields also exist, but decrease significantly from the 5th century onwards. The standard armor of a warrior at the end of the 4th to the 6th century in Sambia was the **dagger knife as a close combat weapon and the lance and spear for long-range combat**. Shields are no longer found in the graves, only dagger knives and spears, but the **horse**, whose importance is evidently increasing, is found as standard. From the fact that, in contrast to Sambia, shield bosses are found in the graves in the surrounding areas, it has been concluded that there was a fundamental change in warfare at the end of the Migration Period and also a change in society in Sambia [162]. The almost complete absence of bows and arrows and axes is also striking. It is assumed that these peculiarities are due more to the burial rite than to extreme deviations in economic and warfare behavior. The standard arming of the

dead with only dagger knives and spears/lances could indicate a special **symbolic significance of these weapons as status symbols** [159]. Particularly wealthy weapon graves from the 6th century onwards indicate the establishment of a **warrior aristocracy** that lasted until the Old Prussian transition period from the Migration Period to the Early Middle Ages. Men's graves equipped with weapons and horses are still typical in this period and can also be found, for example, in Kaup-Wiskiauten, where rich equestrian graves are filled with Frankish swords, Scandinavian horseshoe brooches, and Baltic crossbow brooches. These inventories suggest **polyethnic inhabitants** of at least the trading settlements and also point to a **highly organized society**.

The importance of the Sambian Peninsula as a source of raw materials in amber trade over a long period of time might be reflected, at least in part, in the unusually **high density of strongholds**. For the Sambian Peninsula, the large number of approximately 90 hillforts [152] seems to be a characteristic element of the settlement landscape [148]. Until recently, the chronological classification of the vast majority of fortified sites in Sambia was a major research desideratum due to a lack of research and poor publication status [152]. This situation has improved significantly over the last decade and a half thanks to extensive research programmes [148, 163–165]. Using minimally invasive, motor-driven core probes [166], the $^{14}$C-based dating of a total of nine hillforts in the northwest Sambian Peninsula was achieved between 2014 and 2018 [148]. The results lead to a significantly altered perspective on this group of monuments, and consequently, the reevaluation of entire settlement chambers. According to the latest findings, hillforts in the region have been constructed since the Late Bronze Age, during the declining Bronze Age and Pre-Roman, Roman and Post-Roman Iron Age, when the region belonged to the West Balt cultural circle or West Balt barrow culture and Dollkeim-Kovoro culture [154, 159, 167]. In the Late Bronze and Pre-Roman Iron Age the area was influenced by the declining Lusatian culture and Pomeranian face urn culture, with hillforts in the Lusatian culture region potentially serving as models for fortifications in the area [148]. The distribution of hillforts in the Early Iron Age (map on page 254 of [168]) suggests that the Baltic amber trade played a role in their development, with ideas and knowledge of hillfort construction possibly migrating through trade routes [164]. However, hillforts are a widespread phenomenon across the entire Baltic Sea region, appearing as early as the Bronze Age [169]. Especially new is the chronological classification of many hillforts in the first half of the first millennium AD [148]. The Dollkeim-Kovrovo culture of the Roman Iron Age and Migration Period, defined by numerous richly equipped burial sites and other criteria [154, 170] (see above), was previously not thought to be associated with hillforts. Traditionally, beside some pre-Christian evidence [171–173], the main phase of hillforts was placed in the declining Migration Period, or at the transition to the Early Medieval Prussian culture of the Viking Age, probably influenced by Wulfstan's description (see above). However, this period appears to be a period in which very few $^{14}$C dates were found, i.e. in which there seems to have been **less activity on the ramparts** than in any other period (see Fig 5 in [148]). Surprisingly, the Viking Age thus appears to have been 'peaceful' at first glance. However, this is only an apparent contradiction. From Wulfstan's description we can conclude that there existed a four-tiered social pyramid with a group of equal-rank commanders of strongholds at its top [174]. The Prussians kept this organization and sustained the **de-centralized power structure** avoiding changes that took place around them. The conditions around 900 were not so different from those around 1200: aside from the politically independent commanders no central power existed as a permanent institution, it was a so-called 'stateless society' [174]. They had **no stable territorial organization** that would be controlled by some central power. Their strength was the lack of a central power that could be attacked and defeated. Such a strategy appeared to be very successful during the long resistance against their neighbors, who were stronger and

better organized [175]. The method used to date stronghold activity phases is based on the $^{14}$C dating of **fire horizons** in the ramparts [166]. In the vast majority of cases, these horizons were probably formed during the **violent destruction of the strongholds**. In this respect, the dating primarily reflects periods with escalative moments that led to the use of violence (**fire catastrophes or attacks**). The possibility of simple cooking fires, which were later deposited with the surrounding sediments in the ramparts for rebuilding or repair [148], theoretically exists but we consider it negligible due to the relatively large number of $^{14}$C-dates (110 in total according to [148]. According to this understanding, more stable periods, possibly due to the absence of a central power described above, we would rather read as the result of a successful de-escalation strategy over a long period of time.

For the period of the Late Iron Age (8$^{th}$ century to the conquest by the Teutonic Order in the 13$^{th}$ century), *Siedlungskammern mit Burgherrschaften* ("settlement chambers with central power": a number of open settlements in the vicinity of a stronghold, in which a ruling authority controls these settlements) were reconstructed, which were separated from each other by 'holy forests' and **border forests** [176]. Each settlement chamber had one or two strongholds, either as residential stronghold in the center or as **border stronghold** in the forest or on the edge of the forest. The chieftains were supposed to have resided in these strongholds. With regard to this organization, Christian Lübke [174] draws a parallel to the *civitates* of *Geographus Bavarus* in the Slavic settlement area, which can be identified with settlement chambers whose most important topographical and strategic element was a fortification. Timo Ibsen [148] has combined the new and former available chronological information with a GIS-based kernel density analysis, which shows for the western part of the Sambian Peninsula seven micro-regions with a total of 33 hillforts. While there are chronological gaps in the occupation of individual hillforts, from the Late Bronze Age to the Middle Ages, almost all micro-regions that can be provisionally regarded as settlement nodes were **continuously equipped with at least one hillfort** (Figs 3 and 6 in [148]). The location of a stronghold can provide clues to its function. Lowland strongholds in swamp areas, for example, are interpreted as **refuge strongholds**. They often have no or only a thin cultural layer. The proximity to the coast and to waterways suggests a geographical relevance in terms of traffic. This can be seen particularly clearly along the Pregel and Deime rivers in the 9$^{th}$-12$^{th}$ centuries. The fortifications located along these rivers were developed as **control points** later during the Teutonic Order period [152]. With regard to the location of the strongholds, Ibsen [148] comes to the conclusion that, apart from the existence of a suitable terrain, the choice of location must have been dependent rather on superordinate factors, such as the location to communication routes and infrastructural connections, or other strategic aspects, such as political organization and the manifestation of power or even cultural identity. Unfortunately, those aspects have not yet been sufficiently investigated. Most important is probably the division of the landscape into demarcated settlement nodes, to which the spatial and temporal distribution of the hillforts seems to correspond [148].

The Teutonic Order used a strategy of confrontation during its conquest. All the **longitudinal ramparts** in Sambia date from shortly after the conquest. Probably in response to several Prussian uprisings, the Teutonic Knights erected these longitudinal ramparts for their protection (**defense**), but at the same time the ramparts also marked the **borders of the conquered territory** [152].

Sambia is a region that has always been of particular importance due to its rich amber deposits. This is reflected on several levels. Despite the sometimes poor state of research and a number of uncertainties, a whole series of indicators for escalation and de-escalation processes of conflicts can be identified, which can be related to the E–D Pyramid, cf. Fig 9. Very few indicators are effective over the entire long period from the Late Bronze Age to the Middle Ages.

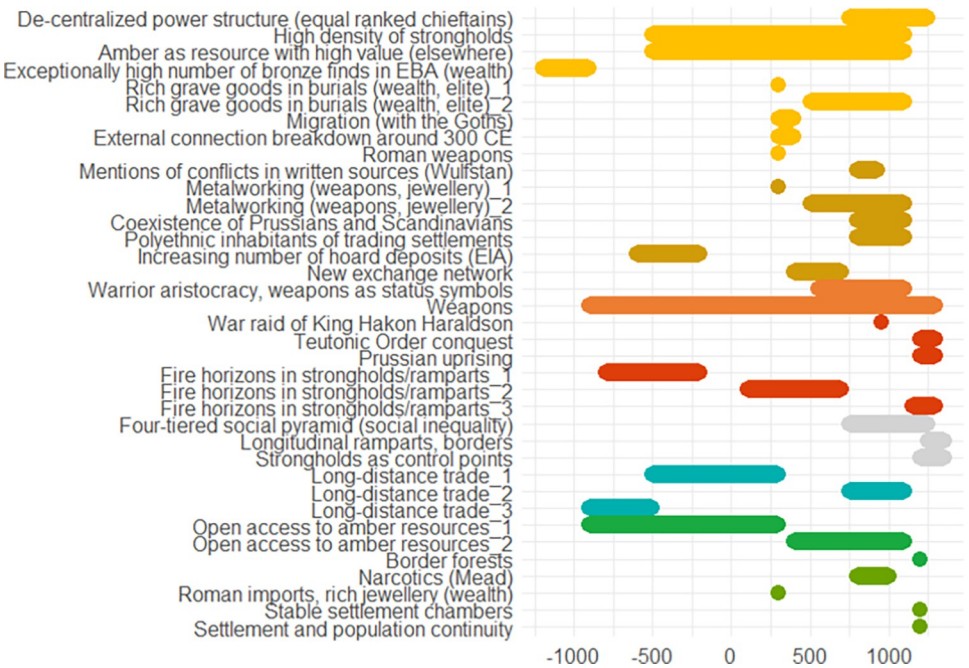

**Fig 9. Approximate chronology of the indicators described in the Sambia case study.**

The essential indicators of conflict potential are: the availability of rich amber deposits, a high density of fortifications and the clear presence of weapons. The majority of the other indicators characterize individual periods in which de-escalation of emerging conflicts was sometimes more, sometimes less successful. Overall, however, there is evidence of relatively high continuity and stability over long periods of time, not broken even by recurring escalations. The decentralized power structure may have made a fundamental contribution to stabilizing the situation.

## Rome (Early Republic-Late Empire): Trade policies

This case study deals with economic policies in the Roman Republic through the Roman Empire and is motivated by the unique possibility of combining historical and archaeological finds to form a comprehensive picture of this period and region. **Law texts** and **policies** are something we can hardly ever grasp from archaeological material, which makes the contribution of historical writings much more important for that aspect of conflict management. Needless to say, the history of Rome offers other kinds of conflict than economical ones, but here (as elsewhere in this paper) we are more interested in drawing an example that lends itself well to analysis than trying to offer a comprehensive history.

To understand conflict potential within the Roman economy, we must look at the policies that have been put in place by the Romans themselves. Several historical sources give some hints on a general **economic strategy** that can lead to or diminish conflict potentials. **Trade** itself is already an indicator of cooperation, regulation, and negotiation. All of these act as de-escalation strategies according to the E–D Pyramid. Due to the nature of negotiation and regulations, however, conflicts can also be created when certain parties are favored over others in economic affairs. The keywords on the escalation side of the pyramid would be **conflict potential** and **interfering interests**—factors associated with all trade or exchange operations among people. Fig 10 shows timelines for all indicators pertaining to this case study.

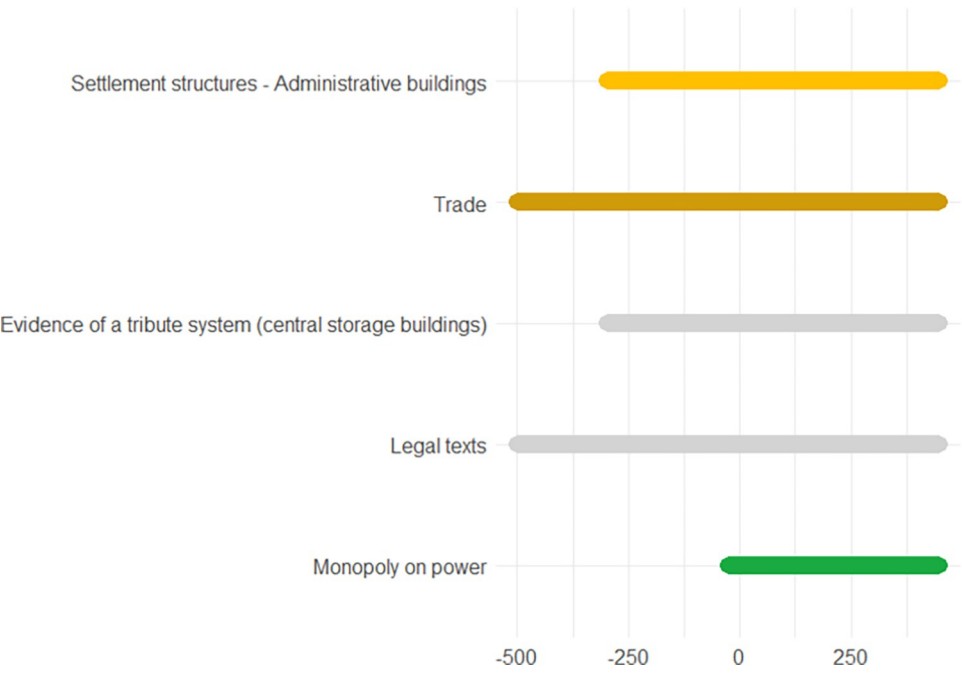

**Fig 10. Approximate chronology of the indicators described in the Rome case study.**

Several historical passages underline that inequality was a known problem in the Roman economy of the early Empire. The following case study will present examples that allow us to reconstruct certain trade policies in the Roman Empire from 50 BCE to 150 CE. This period is of special interest as conflict in the entire Mediterranean region prevailed in pre-Augustan times, while political changes led to a more stable government after Augustus took power in 31 BCE. Wars and battles between the contenders to power that were fought all around the Mediterranean subsided, and external conflicts could also be settled, which led to the long period of peace in the Roman Empire, referred to as the *pax romana*. By focusing on this period of the Roman world, we can investigate both conflict and conflict resolution from the perspective of trade.

We can apply some factors to understand which regions or players were favored or disadvantaged by working with simple parameters that are influential for the economy in general. The interplay of these interfering demands can help us understand hierarchies and power dynamics. These can be applied to the assessment of conflict potential, de-escalation strategies or balancing policies. We will focus on the existence of the three main parameters: **taxes, incoming** and **outgoing customs**. The easiest level to detect in historical literature is that of these parameters and how they are combined for the respective regions of the Roman Empire. Others are rather less easy to determine, such as for example taxes or price settings.

Modern literature on the topic of Roman trade policies tends to create large narratives from the rather sparse evidence present in the primary literature. In addition, little research has been done on the topic since the publication of De Laet in 1949 [177], a publication which can be considered the main authority on the *portorium* (customs) in the Roman Empire. Because of this, we will present the key quotes from historical sources in order to facilitate the evaluation of their validity and applicability to the questions posed. We only focus on investigating if the parameters relevant to this study are either in place or not, with the ultimate aim of judging conflict potential within the framework of the E–D Pyramid.

The few mentions of changes to the system are likely to imply that everything not mentioned continues to function as before. For this reason, we need to start the case study before

the period of investigation. The first mention of tolls and taxes can be found in Livy when he describes the downfall of the Macedon Empire, which roughly dates from 219–167 BCE. Several measures taken by officials of the Republic during this time are enumerated. Among these, it is mentioned that "various tolls were initiated" (40.51.7 in [178]).

In one of his later books, Livy again mentions the same two statesmen acting between the time of the 2nd Macedonian War (200–197 BCE) and the Peace of Apamea (188 BCE), when they established one guiding principle of the Roman economy, namely the special status of Roman and Italian allies who needed to continue to be exempt from harbor dues (38.44.4 in [178]).

The exemption of Romans and Italians from local taxes is an essential hint given in the literature that will be underlined further by the establishment of a law regulating this privilege, the *lex Caecilia de vectigalibus* in 68 BCE by the praetor Q. Metellus Nepos. The law mostly deals with removing the *portoria* from Italian ports and not the province in general as this has caused problems to the domestic economy, furthering the special status of Italy to actual law (37.51.3 in [179]).

Less than a decade later, in 60 BCE, we have another mention of the exemption of taxes and customs in Italy, this time by Cicero. He speaks about the unrests of the conservatives due to the fact that no "home tax is left except 5 per cent, now that the customs duties have been abolished" (2.16 in [180]).

Just a few decades later, according to Suetonius [181], Caesar reinstates the customs for Italy again, imposing taxes on the importation of foreign goods. This move was motivated by the extreme spending in the preceding civil wars and the battle on excessive living standards in the Roman society. For somewhat similar reasons, even more taxes are imposed on the Roman citizenry under the second triumvirate in 39 BCE (48.34.2 and 48.34.4 in [179]).

Clearly, the troubles of the late Republic took their toll on the treasury of the Roman state, eventually leading to an increase in taxes of an unknown kind.

By the end of the Republic, then, Italy was to a large part exempt from taxes and customs on material coming into or leaving the country, a practice that had been carried over from much earlier times. That exemption only changed slightly when Caesar again raised taxes on goods brought into Italy. This left the custom-free export as the only main advantage that Italy had over others. Little else changed afterwards except for the organization of the custom provinces and their borders under Tiberius, where the rates to be paid on crossing the tax regions had been fixed, as can be seen in Table 3. The information in the table has mostly been

**Table 3. Tariff zones in Tiberian times according to [177].**

| Tariff zone | Percentage ad valorem (of the value) |
| --- | --- |
| Sicily | 5% |
| Africa | 5% |
| Asia | 2.5% |
| Gallia | 2.5% |
| Italy (west coast) | 2.5% |
| Iudea | 2–2.5% |
| Spain | 2% |
| Illyricum | 2% |
| Egypt (certain nomoi excluded) | 2% |
| Italy (east coast) | 2% |
| Achaia | Unknown |
| Cyrenaica | Unknown |

reconstructed from epigraphical sources. The reasons behind the varying tax rates for the regions are not clear and thus subject to speculation.

Several more mentions of tax and toll exemptions can be found in the historical sources but no real adaptions took place until the Late Empire. One of the last ideas of changing the customs system came from Nero, who had been interpreted to want to introduce changes in order to make himself more popular with his subjects. This was not welcomed by the senators of Rome and thus never implemented (13.50–51 in [182]).

To summarize, what were the parameters that could create conflict in the early Roman Empire? Fig 11 gives an overview of the differences between the Roman provinces and the provinces of Italy. In total, not many such differences can be seen: the only one being that exported good were not subjected to customs when they came *from* Italy. For all other regions, this is not the case, giving Italy an advantage in the price-setting of their goods. We cannot see exceptions like this for any other place, indicating that Italy's products and thus its economy were somewhat protected. The Roman economy did not take it so far as to also boycott or tax

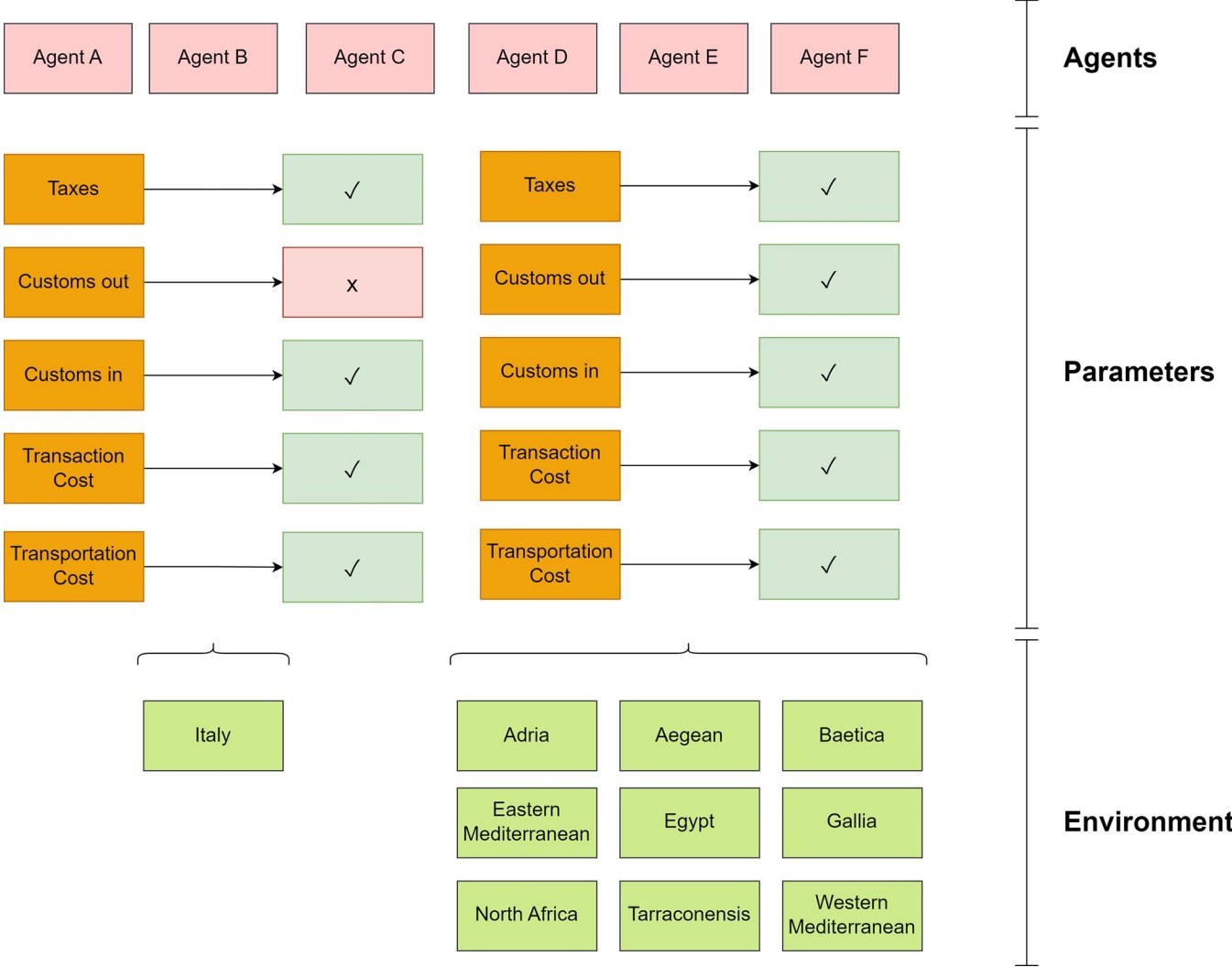

**Fig 11. Historically evidenced concept of trade in the Roman Empire showing the economic parameters in place for Italy on the left side and all other regions on the right side.**

outside products more than others, which resulted in a two-tiered hierarchy in the Roman economy: Italy is the favored and protected region, and all other provinces are treated differently, with no discernably hierarchy among them.

Turning back to the original question of how economic policies in the Roman world were indicators of conflict potential, negotiation, and regulation, we can clearly see that there was a conflict: one region was preferred over all others. Yet apparently, this did not escalate to the point where it was a major threat to the stability of the Empire. After its foundation in 27 BCE and the connected civil wars, revolts were still a common occurrence in the provinces but were mostly put down rather quickly. Some examples include the Jewish revolts in 4 BCE and 6 CE, the Bellum Batonianum in 6–9 CE, the Battle at the Teutoburg Forest in 9 CE, the Boudica revolts from 60–61 CE, the First Jewish War from 66–73 CE, the Year of the Four Emperors (68–69 CE) and the Bar Kochbar Revolts from 132–136 CE. We could propose many reasons why this situation did not escalate further, the most likely being the imperial character of Rome and the economic and military power of the capital itself. In the end, this was an Empire and not a voluntary union of several regions of the Mediterranean. The power of Rome was maintained, but this surely was not to the disadvantage of the provinces, which did, in fact, receive something in return: access to the trade network and redistributive measures overseen by the Roman government. This assumption of reciprocity can be further supported by archaeological data, such as the **foundation of Roman colonies in the provinces** while indigenous settlements co-exist, the **system of administration** put in place in the provinces that did not always overrule the local customs, **law documents** such as metal tablets found in Spain, and **warehouse distributions in the provinces** that might shed some light on **tribute systems** that had been in place. Yet, we must also focus the role of military power of the Roman state that helped keep the system stable. Unequal treatment in trade policies of regions in the Empire may even have partly led to de-escalate the Italians' feeling that the provinces were not giving Italy anything in return for protection and integration into "their empire". Yet, the concept of **structural and military power** that poses a direct threat to those who do not accept the regulations of the Roman state, plays a large role in all of these measures.

## Hedeby (late 8th century–mid 11th century): Long-distance trading and de-escalation strategies

This case study deals with the most important Viking Age (late 8th century–mid 11th century AD) trading center in Northern Europe. The area under consideration comprises the so-called Schleswig Isthmus at the southern foot of the Cimbrian Peninsula and stretches from the North Sea to the Baltic Sea (Fig 12). The choice of this case study is motivated by the border location of the site, the supra-regional importance that it held for around 250 years, its high potential for involvement in conflict, and the de-escalation strategies that were evidently effective over a long period of time, serving to maintain its function as a central trading center. The following indicators were identified during the analysis: trade, destruction, relocation of the site, rampart, place names for warrior groups, signal fires, ship burning, sea barriers, cultural differences, parcelling, central authority, merchant association (cf. Fig 13).

From the beginning of the 9th to the middle of the 11th century, the western shore of the Haddebyer Noor at the south-western end of the Schlei was the most important transhipment point for goods of all kinds in northern Europe as well as a center of specialized crafts, a place of representation of power, and the starting point for the Christianization of Scandinavia. Here, at the narrowest point of the Cimbrian Peninsula, water and land routes from all four points of the compass met: In the immediate vicinity was the only gate of the Danework (Old Norse: *Danavirki*), through which an important north-south road connected Scandinavia with

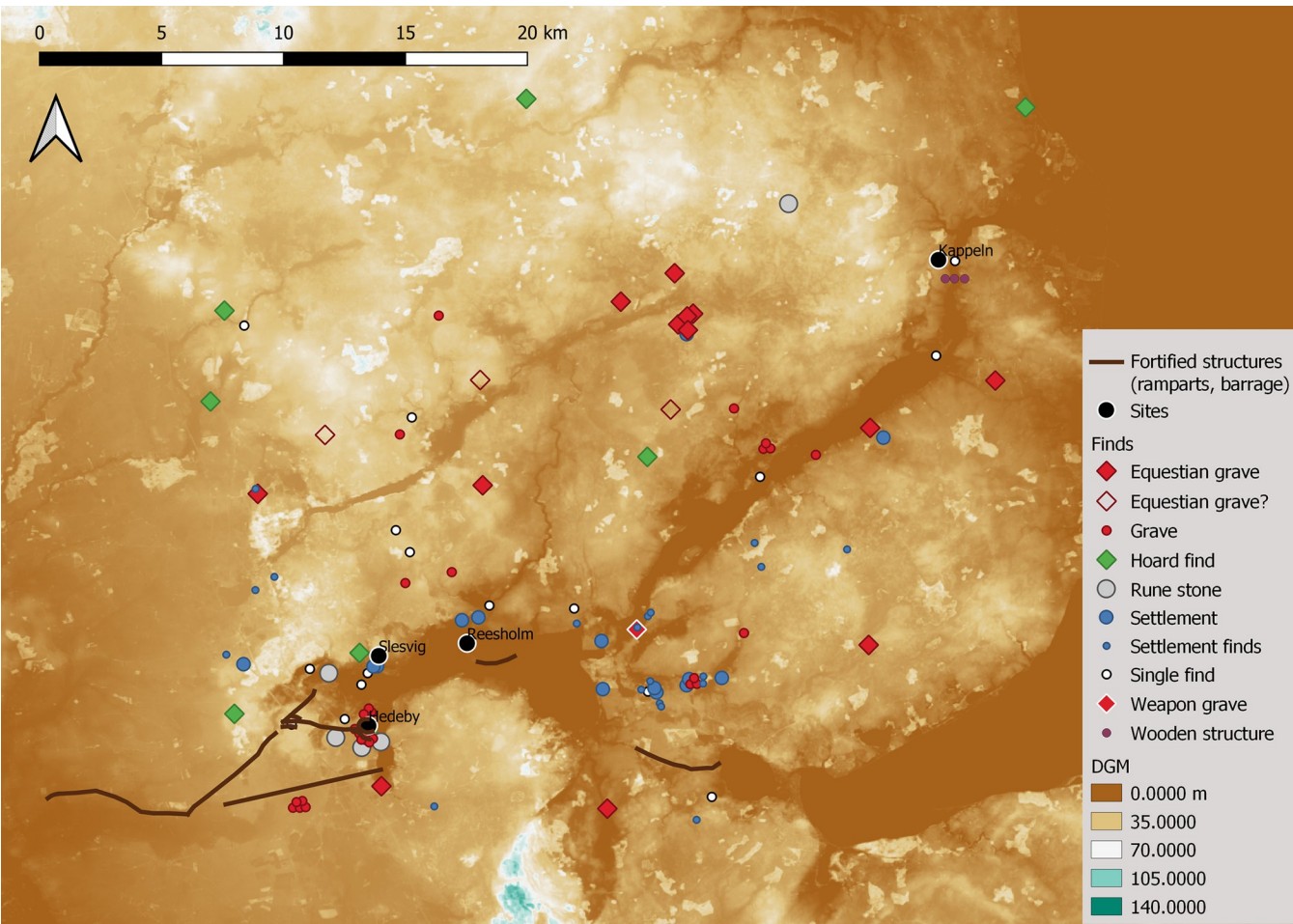

**Fig 12. Hedeby at the inner end of the Schlei-Fjord.** Based on Fig 41 in [183] and Fig 1 in [184].

the continent. Ocean-going ships from the Baltic Sea reached the port of Hedeby via the Schlei and, under the protection of the Danework, goods came over land that had been unloaded at Hollingstedt on the river Treene by ships that were able to get there from the North Sea via the river Eider. Like many other early medieval trade centers, called "emporia", Hedeby was located in a border area. The border to the Franconian, later German, Empire, ran only about 20 kilometers south along the Eider. The Limes Saxoniae, which separated the Slavs from the Saxons, was a day's journey to the south-east.

In the 10th century, Hedeby was around 25.5 hectares in size and, in large parts, densely built. Its buildings and several cemeteries testify to more than a thousand inhabitants, at least during the summer season. It was not until the second half of the 10th century that a **rampart fortification** was built, which was connected to the Danework. Prior to this—according to the current state of research—the settlement was unfortified on the land side (see [183] and [185] on the geostrategic and political location and settlement development of Hedeby). This is somewhat surprising since the potential for conflict was extremely high at this particular location. The city was particularly attractive for predatory raids, revenge campaigns, and political or religious conflicts, not only because of its immediate proximity to the border, but above all because of the regular presence of foreign **traders**, warriors, kings and their representatives, missionary priests and monks, as well as the concentration of enormous material values for the

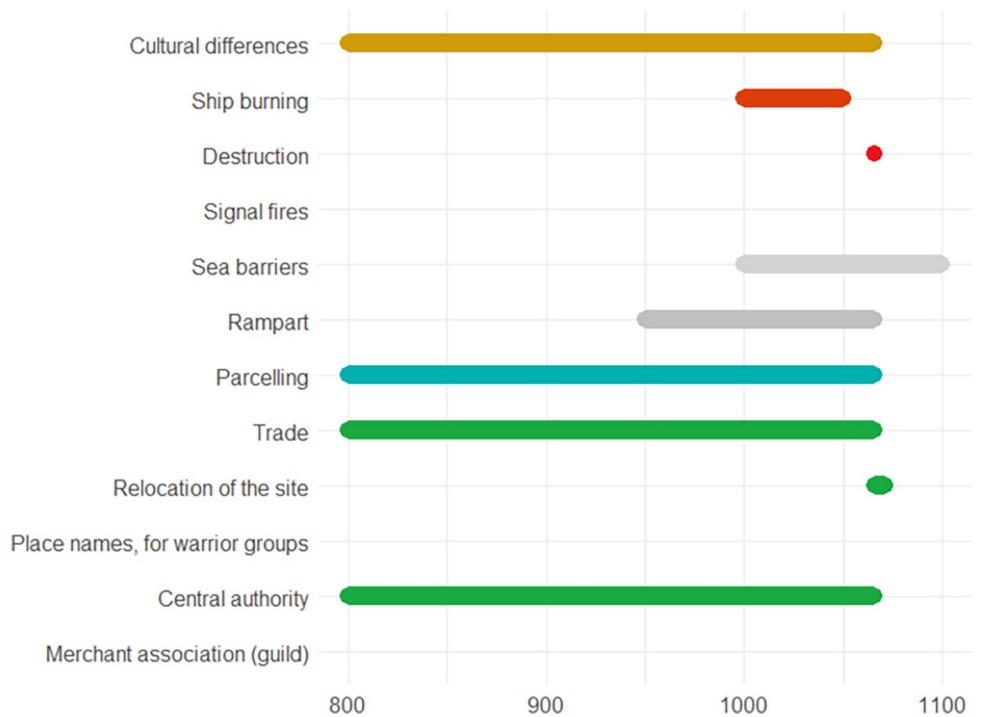

**Fig 13. Approximate chronology of the indicators described in the Hedeby case study.** For three of the indicators dates are not available, thus no timelines are shown in these cases.

time in the form of goods, raw materials (including precious metals), slaves, coins, weapons, and ships.

In addition to the external threat, which materialized several times in Hedeby's history and developed into to raids that ultimately led to its **destruction** and end, the high potential for conflict within Hedeby should not be underestimated. For instance, negotiating a trade deal can easily get out of hand, especially when **cultural differences** may cause friction. Contributing to the sources of differences, the Christian faith had gained a northern foothold in Hedeby. How did this central place manage to function continuously and reliably as a seaport, trading and craft center for more than two centuries despite the potentials for conflict?

Even if there is no evidence of a fortification before the semi-circular rampart was built, Hedeby had not previously been entirely defenseless. It can be assumed that the settlement was founded in the early 9[th] century under the direct influence of a **central power** [183]. This power guaranteed protection through its retinue, as evidenced by rune stones erected around the settlement and elsewhere in honor of warriors who died defending Hedeby. The coins minted in Hedeby were subject to royal influence [186] and reflect the regular presence of the central power.

In the west, the Danework can be attributed a protective function for the land route from Hollingstedt as well as for Hedeby itself through the Kograven. However, like the semicircular rampart, the Kograven was only built in the second half of the 10th century [185]. In the east, Hedeby's protective area extended far beyond the palisades in front of the harbor in Haddebyer Noor and the eastern elements of the Danework (Seesperrwerk near Reesholm and Osterwall) into the surrounding area, as can be seen from **place names** as well as archaeological sites on and in the fjord Schlei. The archaeological sites include the remains of a wooden structure, perhaps a sea **barrage,** south of Kappeln, which probably already existed during the

Viking Age [184], as well as equestrian and weapon graves from the 10<sup>th</sup> century north and south of the Schlei, which may be interpreted as evidence of mounted warriors of the royal retinue. The location of these graves in the immediate vicinity of the Schlei suggests that this retinue was deployed to protect the sea route (for the archaeological sites and the place names see [184]). Just how necessary this was is shown by the discovery of Hedeby Wreck 1, the longest known warship of the Viking Age, which probably was used as a **burning ship** against the landing bridges [185].

The place names *Schnek Wisch* at Lindauer Noor, *Schnickstedt/Schneckstedt* near Goltoft and *Schneckeberg/Schneckenbarg* near Fleckeby are names that can be traced back to the old Norse ship name *snekke*. A *snekke* was a small seagoing vessel, a warship or a 'leding'-ship ('leding' = medieval maritime levy organization) [184]. The name *Rinkenæs* (Low German *Rinkenis*) to the east of the narrows of Arnis means 'warrior cape' in Old Danish [187] and provides a further indication of the presence of a controlling/protecting retinue. The place names *Barnhoe*, *Barnhöi*, *Barnhye*, *Barnhy* and *Barnhö* in Angeln may also have been significant. Their onomastic interpretation points to early **signal fires** that reached far inland from the coast [184]. This communication could have originated from the location of *Warhuje* north of Kappeln. This name appears to derive from Old Danish *warth* 'watch', 'lookout' and Danish *høj* 'hill' [184]. A viewshed analysis showed that several nearby pairs of sites carrying the toponymic element *Barn-* were mutually visible, suggesting that they were part of a chain of beacons [184]. Even if there is a lack of datable evidence, these place names in connection with the horsemen's and weapon graves could be seen as indications of a sophisticated signalling and defense system that required a central organization.

In Hedeby itself, settlement structures provide evidence of an organization of coexistence that must have had a de-escalating effect. Geomagnetic measurements show the planned layout of paths and plots in areas close to the shore and thus close to the harbor, which could also be proven in the archaeological excavations [183]. Although deviations occurred in the course of settlement development, for example when houses were rebuilt at different locations and paths were built over, there is also evidence of continuity over generations. For example, there were plot boundaries over the course of a century, which were renewed several times on the same spot. Perhaps these continuities are indications of larger landholdings or an expression of joint work by several neighbors [188]. Of particular interest is the north-south path running parallel to the shore, along which many plots are oriented. Its uniform appearance suggests that it was not the neighboring plot owners or tenants, but some higher authority that was responsible for its maintenance [188].

As far as can be seen, the **parcelling out of** the settlement area near the shore was continued right up to the wooden landing stages. This means that the landing stages, which were later connected to form large platforms, were probably used privately as extensions of these plots [185]. It does not seem to have been arbitrary where which ship was unloaded and who received which goods or sent them on their journey. This system of parceled-out harbours continued even after **the harbor was relocated** to the north bank of the Schlei in Schleswig in the second half of the 11th century [189].

This ultimately leads to what is probably the most important, but most difficult to prove de-escalation strategy: the organization of solidarity and community through trust and friendship, even over great distances. As recently stated by Siggurdsson, "Viking Age society was a network society. All of Scandinavia was tightly connected by social networks created by friendship and family ties" (page 2 in [190], see also [191]). Long-distance trade in particular required a high degree of trust, especially in the effectiveness of oral commitments due to the lack of written forms [192, see also 193]. It is therefore not surprising that the **beginnings of merchant guilds** can be presumed for Hedeby and Schleswig in pre-Hanseatic times [194, 195]. In turn,

the ports were of central importance for the formation of such communities [196] and Hedeby was a place of de-escalation in the conflict-ridden world of the Early Middle Ages [197].

## Rus' (980–1015 AD): Territory, alliances, and religion

For the following case study, we delve into the early medieval Kievan Rus', a region that underwent a development towards early statehood from the second half of the 10th century onwards. Its territory stretched from the Baltic to the Black Sea during the 980–1015 AD rule of Vladimir. The area was characterized by the coexistence of various locally established groups—such as Slavic, Baltic, Finno-Ugric, and Turkic tribes—but by the end of the first millennium AD it was predominantly influenced by Slavic culture, hence the subsequent discussion of the East Slavic early state. The presence and interaction with various tribes makes the examination of conflict resolution strategies intriguing; moreover, the Rus' were also interconnected with external actors and kingdoms on the international stage. The territory of the Rus' served as a kind of transit zone for many of these actors. The area attracted vibrant trade connections between East and West, as well as North and South.

This case study draws upon the example of the rule of Vladimir to demonstrate the dynamics of foreign policy and intra-societal group interests within the territory. The relationship between Byzantium and the Rus' is particularly suitable for this purpose: although the accuracy of historical written sources, especially the so-called Nestor Chronicles of the Rus', is not always guaranteed, the reliability of the statements can be increased by consulting Greek sources. In the analysis of the Byzantine-East Slavic relationship, only selected aspects will be addressed, but enough information will be provided to enable a first impression of the complex social and political landscape of early medieval Rus'.

The relationship between Byzantium and the Rus' was initially ambivalent: from the mid-9th century to the mid-10th century AD, there were signs of both cooperation and potential conflicts between them, often intertwined. For example, in 860 AD, a large-scale attack on Constantinople was reported, in which the Rus' were involved, among others (**threat: physical violence**). The incident ended with the first **peace treaty** between both parties [198]. Few **treaties** between the Byzantines and the Rus' have been preserved, but legal-historical analyses of those from 911 AD and 944 AD suggest that both treaties regulated the norms of interaction between them [199]. According to [200], these legal aspects concerned the behavior of the Rus' on Byzantine territory and regulated the legally binding handling of slave trade, theft, and trade issues. The fact that the treaties probably only applied on Byzantine territory, and that the treaty of 944 AD was a result of renewed attacks by the Rus', may suggest that the Rus' posed a serious threat, which the Byzantines attempted to control through treaties and **laws**. Despite the animosities, there are reports of Rus' who regularly served on the side of the **Byzantine military services**, although for payment, such as during the so-called Lombard campaign (934 AD), joint attacks on Crete (949 AD, 961 AD), or against the Hamdanids (955 AD).

From this ambivalent relationship it can be inferred that the main interest of the Rus' in dealing with Byzantium was often guided by **economic benefits**, whether in the form of payments or trade privileges. The relationship between both actors finally stabilized during the rule of Vladimir. When the Byzantine Empire under Emperor Basil II (who ruled 976–1025) faced political revolts in Anatolia, the emperor requested military support from Vladimir and his Varangian warriors around 987 AD. The conditions of this **military alliance** are documented in writing and, according to [201], testify to the exceptional political predicament in which Basil found himself: Vladimir demanded, in return, to marry Basil's sister Anna. This demand was a sacrilege, as purple-born Byzantine princesses were not typically given to foreign ruling dynasties: "Marriage into the ruling house of the Byzantine Empire was a singular

honor, rarely granted, and therefore strikingly significant in that it would bestow such high stature on a new addition to the Christian world" (page 8 in [201]). Nevertheless, Basil accepted this demand, albeit expecting three things in return: military support against the rebellion, Vladimir's abandonment of his previous wives, and conversion to Christianity. Ref. [202] describes as part of the agreement not only the personal baptism of Vladimir but also the Christianization of the entire country.

Although Vladimir and his warriors fulfilled their part of the agreement, the Byzantine emperor hesitated to send his sister Anna and her entourage to Vladimir. According to one version of the subsequent events, Vladimir then commanded an attack on the Byzantine trading post Cherson on the Crimean peninsula—presumably to underscore his demands in the treaty (**threat: physical violence 2**) (cf. [201] contra [203]). Apparently, the attacks had an effect, as Anna was subsequently married to the ruler of the East Slavic early state after Vladimir was baptized (**dynastic wedding**). As a gift to his brother-in-law, Vladimir gave Basil the recently conquered territories around Cherson (**cede of territorial claim**) [201].

The commonly accepted date of Vladimir's baptism—and thus the contractual beginning of the Christianization of the East Slavic early state—is the year 988 AD (**religious conversion**) [201, 202]. The dynastic alliance forged through the political interests of the Byzantine court resulted in a restriction on the succession within the East Slavic realm: henceforth, only the descendants of Vladimir and Anna were entrusted with ensuring the ongoing continuity of the royal lineage. Vladimir attempted to establish an **objective regulation of succession**: up till then, the East Slavic rulers had relied on their descent from the Norseman Riurik, who was allegedly summoned by Slavic and Finnish tribes in the 9th century AD to rule over them and establish peace among them [204]. Thus, in the history of Kievan Rus', disputes regularly arose, as evidenced by conflicts among Vladimir and his siblings, who resorted to fratricide in their pursuit of sole rulership over the East Slavic domain. Consequently, after Vladimir's death in 1015 AD, this "Byzantinization" of the succession had to fail, as the exclusion of the older heirs, who did not stem from the marriage with Anna, encouraged their politically motivated murder of the now "legitimate" heirs to the throne [205]. Instead of objectively regulating the succession and protecting the claim to rule against third parties, this succession right thus led to the exact opposite.

For the seat of power in Kiev, there is also evidence of a strengthening Byzantine influence: Martin [201] refers to a Greek population that settled in Kiev after the dynastic wedding (**migration: Byzantine population**). However, whether this mainly consisted of Anna's royal entourage or craftsmen remains unclear. Nevertheless, a strong Byzantine influence and knowledge transfer can be observed in general, which manifested itself at the beginning of the 11th century AD in the adoption of Greek traditions, technologies, and architectural styles [201].

With the Christianization came the **transformation of the architectural landscape** in Kiev: after the destruction of pagan idols and temples, Vladimir symbolically erected the 'Church of the Tithes' on the central hill of Kiev, where it was visible to all residents and visitors of the city (**prestigious building: prominent location**). As the first church in Kiev, it was built by Byzantine workers around 989 AD and completed in 996 AD (**prestigious building: foreign stylistic influences**) [201]. Shortly after Vladimir's death, another church was built in Kiev around 1037 AD, the Saint Sophia Cathedral: following the Byzantine model of Hagia Sophia, it was demonstrably the cathedral that symbolized the center of political life in Kiev. Rulers were crowned here, popular assemblies held, and diplomatic envoys received according to courtly protocol (**prestigious building: seat of power**). Political dignitaries were also buried here.

Although, as Trunte notes (page 189 in [202]), Vladimir's baptism in 988 AD cannot be equated with the onset of Christianization, as there had been contacts with Christianized regions in the East Slavic area centuries earlier, it is all the more remarkable that the population still predominantly adhered to their pagan faith. Moreover, there may have been a significant difference in the perception of the East Slavs between voluntary conversion and forced conversion. There are indications that the forced religious change within the East Slavic population led to considerable tensions and conflicts. The refusal to accept the Christian faith was by no means surprising. Vladimir's father, Sviatoslav, had already decided against adopting Christianity for fear of losing the loyalty of his retinue. Therefore, it is not surprising that the population of Novgorod violently resisted the introduction of Christianity around 988 AD (**religious conversion: resistance**), especially since Christianization involved the desecration and destruction of their familiar pagan idols. The erection of these idols and the construction of temples had been Vladimir's political-religious attempt just a few years earlier to establish an identity-forming belief and value system for the various population groups within the Rus'. In 983 AD, "two Varangians who had become Christians in Constantinople died as martyrs" (page 191 in [202], transl. by the authors) which again speaks for Vladimir's political agenda behind the religious change, rather than conviction.

The emerging potential for conflict necessitated some actions to be taken to decrease intra-societal tensions and potential conflicts. One of these actions concerned customary law, which evolved under Vladimir into a kind of orally established 'catalogue of punishments' to regulate crimes within his domain. Crimes could now be compensated by a monetary fine, which was distributed among the families of the victims, the sheriff, the church, and the ruler [201]. The written fixation of this orally transmitted customary law, the Russkaja Pravda, was finally accomplished during the 11th century by Vladimir's son Yaroslav. With the now **binding jurisdiction**, there was now a regulation to prevent intra-societal unrest (**law: written fixation**) [199, 206].

Instead of the familiar polytheistic idols, Vladimir made use of the Christian tradition of saint worship. Patron saints were adopted in the Byzantine tradition, revealing the idea that East Slavic rulers like Vladimir should probably be worshipped after their death: Poppe refers to the placement of Vladimir's resting place in front of the main altar of the palace church, which speaks for a "veneration of Vladimir in Kiev immediately after his death", contrary to Byzantine tradition (page 405 in [205], transl. by the authors). This procedure has clearly de-escalating features as it offers the population alternative figures for identification through the **veneration of saints and rulers**.

It is also likely that, in a secular context, a conscious decision was made against the Greek language in favor of a liturgical language familiar to the population, namely Old Church Slavonic [202]. Old Church Slavonic was introduced as the obligatory ritual language in the East Slavic early state. The linguistic proximity between early East Slavic and Old Church Slavonic would have had a positive effect on the willingness to accept Christianity, but language standardization usually also comes with a cost of disputes over linguistic norms (**religion: identity-forming**; **familiar missionary language: Old Church Slavonic**).

Another method to contribute to appeasing the population was the regional reign of Vladimir's sons as administrators in the diverse districts of the Rus', which he declared as the earliest centers of the Orthodox Church. Thus, they were supposed to introduce secular administration in the districts as **representatives of the political administration**, protect the Christian clergy, and persuade the local population to abandon their pagan deities.

This case study focuses on the political relations between Byzantium and the Rus', which were initially characterized by ambivalence during the 9th and 10th centuries. Numerous historical events such as attacks on Constantinople, peace treaties, and military alliances illustrate

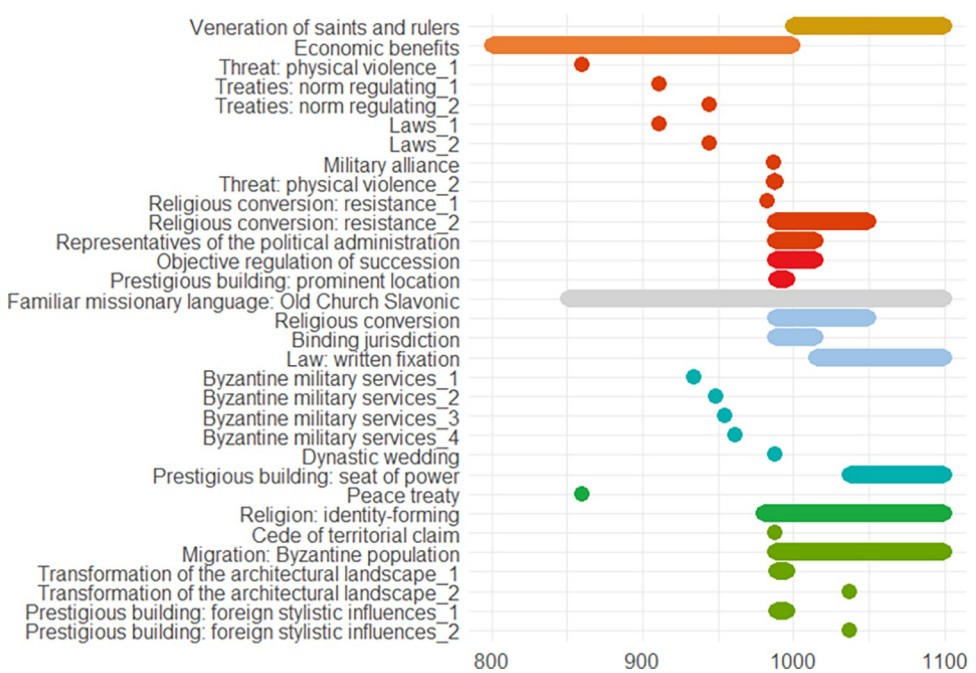

**Fig 14. Approximate chronology of the indicators described in the Rus' case study.**

the potential for cooperation as well as conflict. Especially during Vladimir's rule, the relationship between the East Slavs and Byzantium stabilized as a result of the dynastic wedding with the Byzantine princess Anna. Vladimir's subsequent baptism marked the beginning of the Christianization of the Rus' and led to profound changes in the religious, political, and architectural landscape of his domain: With the adoption of Christianity, it was possible to expand the political sphere of influence while simultaneously establishing an identity-forming basis as a foundation for common values. Despite the success of Christianization, there was resistance from the population. Vladimir's actions to appease the population, such as the introduction of an orally established customary law, a familiar missionary language, and the appointment of his descendants in the earliest ecclesiastical centers in the East Slavic early state, decisively contributed to stability and a reduction of conflict potential.

An overview of the indicators relevant to this case study is given in Fig 14.

## The Teutonic Order (1226–1525 AD): Knights, kings, wars, and fealties

The following case study, based on [207–209], deals with a long and complex conflict in today's Poland and the Baltic states. It played out between the Teutonic Order—one of the prominent crusading orders—and the Old Prussians and later the King of Polonia. Various other entities and conflict parties were involved. The most stable front lines, however, continued over three centuries between the Order and the kingdom of Poland. It certainly goes without saying that within these three centuries, the most prominent antagonistic parties changed constitutional formation, headquarters, personnel, strategies and, last but not least, their aims, making the conflict one of the most oscillating and yet constant conflicts in Europe. The conflict dates back to 1226 and was continued as an armed conflict until 1525. During this period, the conflict partners changed their constitutional formation and networks of allies. Various escalation and de-escalation strategies characterize the conflict, the latter including attempt to solve it through diplomatic means. We consider the following indicators important for the conflict:

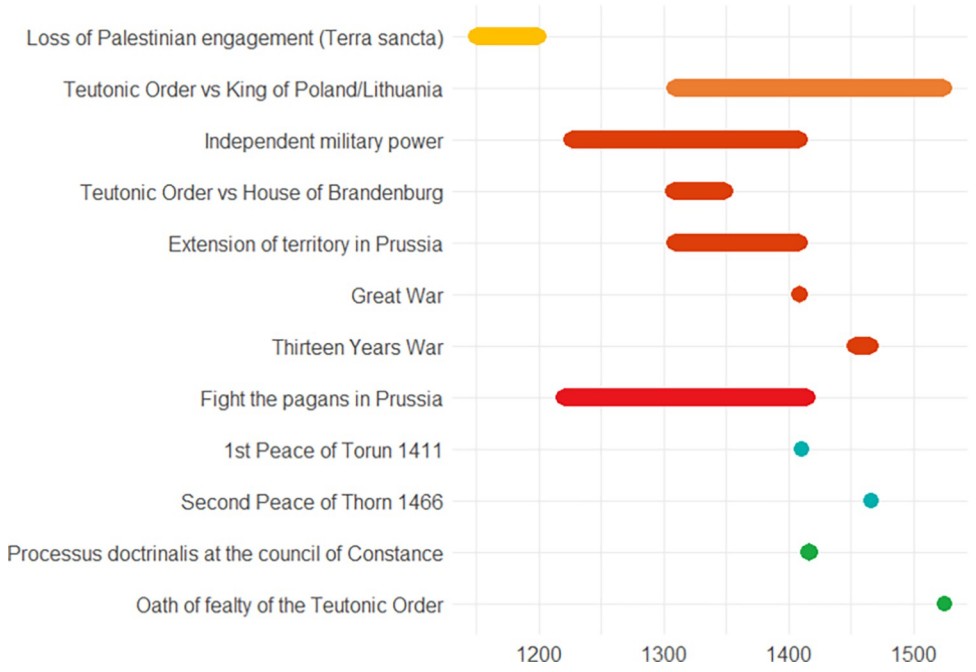

**Fig 15. Approximate chronology of the indicators described in the Teutonic Order case study.**

(1) loss of Palestinian engagement (*Terra sancta*); (2) fighting the pagans in Prussia; (3) independent military power; (4) Teutonic Order vs. House of Brandenburg; (5) Teutonic Order vs. King of Poland/Lithuania; (6) Extension of territory in Prussia; (7) Great War; (8) 1st Peace of Torun (1411); (9) *processus doctrinalis* at the Council of Constance; (10) 13-years war; (11) Second Treaty of Thorn (1466); (12) Oath of fealty of the Teutonic Order. A graphical overview of indicators is provided in Fig 15.

In 1226, the Polish Duke Conrad I of Mazovia asked the Knights of the Teutonic Order (Order of the Brothers of the German Hospital of St. Mary in Jerusalem) to help him fend off attacks by pagan Prussians (*Pruteni*) from the east. The order was originally founded in 1198 as a community of brothers who cared for a hospital in Jerusalem. One of the requirements for joining the order was that the knights were of noble birth and that they took a vow of celibacy, renounced their private property, obeyed their superiors, and fought against pagans. Alongside the Knights of St. John, the Knights Templar, and the Knights of Malta, the Teutonic Order was one of the internationally operating orders of knights. After the loss of Jerusalem (**Loss of Palestinian engagement** / *Terra sancta*), the Order was looking for a new field of activity, so Conrad of Mazovia's commission **to fight the pagans** was very welcome. In return for the fight against the pagan Prussians, the duke gave the Teutonic Order the Kulm Land and all future territories conquered by the Order in Prussia ('Kruschwitz Treaty', 1230), which the Order had confirmed by Emperor Frederick II and Pope Gregory IX, serving as a **diplomatic and contractual base** for the expansion of the order. This was the basis for the expansion of the Teutonic Order in Poland beyond the Kulm region and its growth into an **independent military power**.

After the Old Prussians were defeated, the Polish king Władysław Lokietek asked the Teutonic Order to help him defend the city of Gdańsk against a military **campaign by the Duke of Brandenburg**, who certainly was a Catholic Christian ruler. However, after occupying Pomerania and the fortress and city of Gdańsk in 1308, the Order did not hand over the territories to

the Polish king. This was one of many subsequent disputes between the Polish crown and the expansive state of the Order. The land gained was also of major strategic and economic importance for the Order, as it meant that the entire lower reaches of the Vistula, including Gdańsk, were under its rule. In 1309, the change of focus was completed and the Teutonic Order dissolved its headquarters in Venice. The new center became Marienburg Castle on the Nogat River, which was built for this purpose. The state of the Order in Prussia reached the peak of its **expansion** at the turn of the 14th and 15th centuries thus becoming an **imminent threat to the Monarchy of Poland**. Gotland, the Dobrinerland, Samaiten, the Neumark and some northern parts of Mazovia belonged to its territory for shorter or longer periods of time. As there were still Old Prussians to be Christianized through the Order, it was an attractive opportunity for noblemen to fulfil a military mission in combination with a pious cause. Therefore the military staff of the order was supported by hundreds of noblemen from all over Europe—mainly Rhineland, but also France, Spain, England or Hungary, who wanted to defend and expand the frontiers of Christianity. In the course of the conflict, parties to the conflict were formed. On the Polish side, there was a differentiation between the monarch and the state. The monarch led both the Kingdom of Poland and the Grand Duchy of Lithuania since the Union of Krewo in 1386. The kingdom of Poland formed as a social and political entity under the concept of the *Corona Regni Poloniae*. This entity existed independently of the dynastic interests of the ruler. The integrity of the crown's rights and lands should be emphasized, which was expressed in the kings' oaths of allegiance, which committed them not to diminish the existence and to regain the lost territories.

At the beginning of the 15th century, the situation could no longer be pacified by diplomatic means: The conflict between the two powers, which had developed in their respective constitutions on the basis of very different fundamental ideas, could only be resolved by military means. 1409/11 saw the so-called '**Great War**', which even the rulers of the Empire, Bohemia and Hungary, who were called upon to arbitrate, were unable to stop. The military phase of the conflict between Poland and the Teutonic Order culminated in the Battle of Tannenberg (Polish: Grunwald) in 1410. The army of the Teutonic Order, which included knights from other western nations as well as many Poles, fought against a Polish-Lithuanian army led by the Polish King Władysław II. Jagiełło and the Lithuanian Grand Duke Vytautas (Witold). The battle ended in defeat for the Order. The defeated Order concluded the so-called **First Peace of Thorn/Toruń** with the Polish-Lithuanian Monarchy in Toruń on February 1, 1411, which fixed the *status quo* from before the war. The reason for the strong consideration of the Teutonic Order's interests was that although the Order could not face the Polish-Lithuanian king with equal political and military weight, the Order had legal titles that could not simply be overridden. Therefore, in the subsequent arbitration proceedings before King Sigismund and Pope Martin V, attempts were made to obtain these legal titles, but without success. Now the conflict was no longer carried out through weapons but through diplomatic and processional means. The fierce dispute before the Council of Constance between the representatives of Poland and the Order with the help of legal theory in the so-called *processus doctrinalis*, i.e. with the help of science, above all canon law, should also be seen in this light. Politically and militarily, the influence of the Teutonic Order diminished considerably. After the **Second Treaty of Thorn/Toruń** (1466) following the 'Thirteen Years' War' against the cities and estates of the state of the Order allied with Poland (Prussian Confederation), the Order also had to submit legally to the Polish crown. The Order lost Pomerelia and the western half of Prussia with Marienburg Castle, Danzig, Elbing and Thorn. Almost 60 years later, in 1525, the last Grand Master of the Teutonic Order, Albrecht von Hohenzollern, swore an **oath of fealty** to the Polish king as a secular prince on Luther's advice and, in the course of the Reformation,

transformed the state of the Order into a hereditary secular duchy, which fell to Brandenburg in 1618.

## Bohemia (c. 900–1946 AD): Languages and nations

Bohemia as a geographical space has been a **multilingual area** for almost a millennium. In the 10[th] century it can still be considered a more or less monolingual space with Western Slavic dialects being spoken in a continuum. First steps of Christianization were undertaken in the 9[th] century already using Old Church Slavic and the Byzantine rite as the vehicle of the Moravian Mission of Cyrill and Methodius. Obviously this South Slavic language was not perceived as "foreign" in the Western Slavic area, since it was at the time still highly intelligible with Western Slavic varieties. Starting in the 10[th] century, after the fall of the Moravian dukedom, Bohemia turned to the Western, Roman variant of Christianity (diocese of Prague since 973) and by doing so imported a new liturgical language: Latin. Unlike Old Church Slavic, Latin was not intelligible to the population. It thus became the visible (or rather audible) **feature of an elite group**, especially since it was initially connected to reading and writing. There seems to have been resistance to the adoption of Latin Christianity: the killing of Duke Wenceslaus (Germ. Wenzel von Böhmen, Cz. Svatý Václav, by his brother and successor Boleslav in 928 or 935) shows signs of a pagan **revolt**—or at least an attempt to avoid Latin Christianity.

The introduction of Latin Christianity thus made the social stratification of the Bohemian population linguistically more noticable. Latin as an elite language certainly contributed to the widening gap between clergy and (educated) nobility on the one side and the "ordinary" population on the other. Adopting a "foreign" liturgical language in this respect certainly bore a high potential for social conflict.

To a lesser degree, taking over Christianity in its Western form also included the migration of speakers of some variant of German to Bohemia. The first bishop in Prague (named Thietmar) was sent from the archdiocese Mainz and was a speaker of German. Monks who would settle the first Bohemian monastery (Břevnov, founded 993) came from the Bavarian monastery in Niederalteich. Starting in the 12[th] and continuing throughout the 13[th] and the first half of the 14[th] century the influx of German speakers rose exorbitantly, when rural settlers were called into the country to develop sparsely settled or uninhabited regions, mostly at the mountaineous periphery of Bohemia (as part of the so called High Middle Age Land Consolidation). Through traders, but also through the formulation of city rights, which were often based on German examples (e.g. the city rights of Magdeburg or of Norimberg, thus forming legal **networks**) the German language also intruded into non-rural towns and cities. In newly founded towns (**urbanization**) German immigrants usually formed a majority [210]. In the 14th century, when the Bohemian king Charles IV (or Charles I as king of Bohemia) became the emperor of the Holy Roman Empire residing in Prague, the input of German was probably intensified in the capital. Charles's chancellor Johannes von Neumarkt even played a role in the standardization of early German. We have a **massive migration** of German speakers to the area. For speakers of Czech, German is not intelligible and clearly foreign.

Thus, by the end of the 13[th] century Bohemia is certainly a trilingual country. Religious and administrational texts are mostly in Latin. Latin is a special case of a language, because there were no native speakers at the time. Instead the native language of a substantial part of the population is German. The majority of the population, however, still speaks Czech. We can therefore identify two different reasons for a language conflict that continued until the 20[th] century. The introduction of Christianity and Latin as its language is primarily a cultural change, a sort of cultural innovation. The introduction of German is (not exclusively, but mostly) the result of migration.

The massive input of German speakers soon provoked **protective statements** by Czech protagonists. The anonymous author of the so-called Chronicle of Dalimil (mid 14[th] century) formulates worries about the fact that noblemen will not teach their children Czech anymore [211]. By the turn of the 14[th] to the 15[th] century the Czech religious reformer Jan Hus appeals to his fellow countrymen to avoid Germanisms. Obviously, at least for some speakers of Czech, the language situation was alarming.

But, interestingly, the strengthening of the Czech language in the Hussite reformation results rather in limitations to the other language in Bohemia, Latin. Czech is used extensively in the religious sphere in the 15[th] and 16[th] century, thus challenging the exclusivity of Latin as the language of the Bible and in theological debate. The oldest known complete translation of the Bible into Czech was composed around 1360 (Cz. Bible Dražďanská). The first printed Bible in Czech appeared 1488 (Cz. Pražská Bible). In the 16[th] century the Bible of Jiří Melantrich z Aventina was a bestseller. The 16[th] century then already saw the first grammars of Czech. The German-speaking population had almost no part in the **Hussite movement**.

Here language becomes a main issue. According to the Catholic Church the Bible was not supposed to be translated into vernacular languages. Using Czech in this context is a clear heresy—and a threat to the power of elite groups. The intrusion of one language into the functional sphere of another thus carries a great potential for conflict. Throughout the 16[th] century the new rulers of Bohemia (since 1526 the Habsburgs) pressed towards a restauration of the old religious and linguistic situation. In the beginning of the 17[th] century the tensions exploded in violence in the **Prague defenestration** and the beginning of the 30-years-war.

The Czech protestants—and together with them the Czech language—lost this war after the **Battle** of White Mountain (1620). The main actors of the protestant upheaval were forced into exile. Now Catholicism and the rights of Latin were restored (Counterreformation). As an example we may take Prague University: The head of the protestant (utraquistic) university— the famous medical doctor Jan Jessenius—was among the 27 persons who were **executed** in Prague after the Battle of White Mountain. The university itself was put under the rule of the Catholic Jesuit order [212]. Even worse, German was now legally put side by side with Czech as the official language of the kingdom in the "Verneuerte Landesordnung" from 1628. In reality German quickly became the dominating language in the administration of the kingdom (linguistic **discrimination**). In Czech historiography the century between 1650 and 1750 is therefore referred to as "The Darkness" (Cz. *temno*). The Czech language was limited to everyday family life. In public, German **dominated** the scene.

Only in the second half of the 18[th] century, during the first phase of the Czech **national revival**, did the Czech language start to become more important again. Compulsory schooling in German was introduced in the 1780s and further limited the significance of Latin in an educational system that had earlier been based on the knowledge of Latin. At the end of the 18[th] century Latin had **lost much of its cultural importance** in society and was restricted to the religious sphere and higher education. But even the sciences were gradually taken over by vernacular languages. The 1760s saw the first courses being taught in German at Prague University (by Karl Heinrich von Seibt). Teaching in Czech started gradually after 1848 [213]. The language question was now reinterpreted as an ethnic "national" problem between speakers of Czech and speakers of German. The protagonists of the national revival demanded equal rights of Czech and German within the kingdom of Bohemia. Again we find an extension of the functional spheres of one language into the functional spheres of another. All attempts to solve the upcoming conflict by creating a bilingual "Bohemian" (neither Czech nor German) identity failed. A first spark of violence finally occurred at the Prague Pentecostal uprising in 1848, when radical nationalists demanded an independent Bohemia. As a symptom of the failure to bridge the widening gaps between the Czech and the German language one may again take

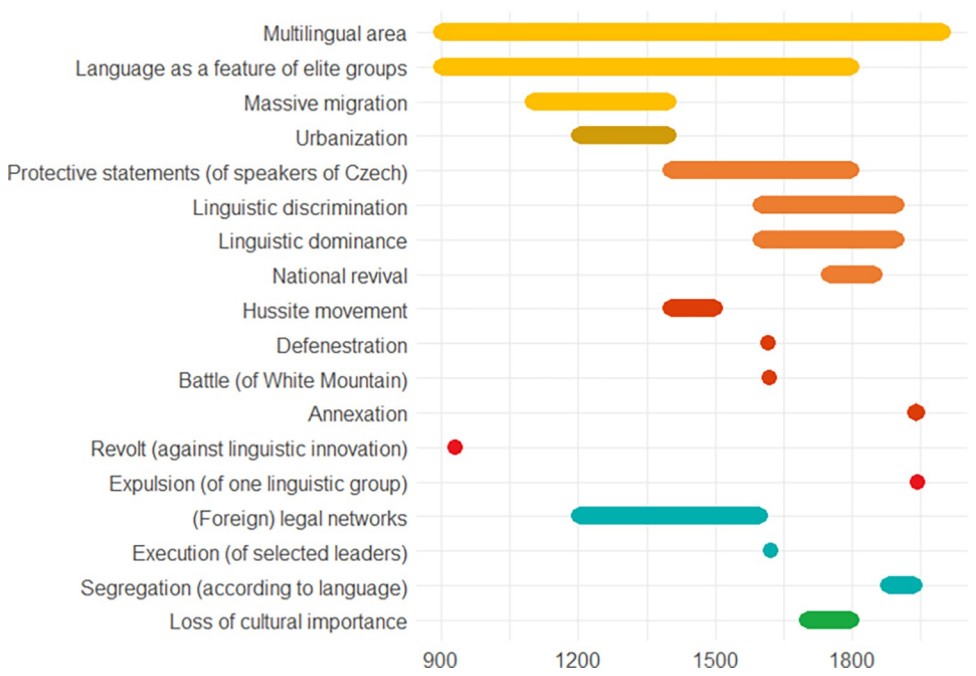

**Fig 16. Approximate chronology of the indicators described in the Bohemia case study.**

Prague University. In 1882 it was split up into two independent institutions—the Czech and the German "Universita Carolo-Ferdinandea" with Czech and German as the languages of teaching (**segregation** of public life). The two separate universities existed until the Czech university was closed by the Nazi regime in 1939 [213]. The claim of the Czech population to a separate political entity was realized after World War I in the newly formed democratic state of Czechoslowakia with Czech (and Slovak) being the official languages of the country. But now the German-speaking part of the population felt linguistically discriminated against, which lead to the **annexation** of the German-speaking parts of Czechoslovakia by Nazi Germany in 1938. At the end of World War II the anti-German counterreaction lead to the final and forceful solution of the language conflict in Bohemia. Practically all German-speaking citizens of Czechoslovakia were **expelled** in 1946.

An overview of indicators is provided in Fig 16.

## Volga Germans (1763-c. 2000 AD): Migration, language, and identity

Through an examination of the history of the Volga Germans, we can illustrate how individual stages of escalation and de-escalation can be applied to an ethnic group with its language and identity, which is why this topic was chosen as a case study. The Volga Germans were German-speaking settlers invited by Catherine the Great in the 18[th] century to cultivate unsettled land around the Black Sea and the Volga Region of Russia.

In 1763, Catherine the Great issued a manifesto granting many **privileges** to German farmers and craftsmen. These privileges included receiving free land, interest-free loans for land development, the right to leave Russia at any time, exemption from military service, freedom of religion, as well as the right to self-governance and the establishment of their own churches and schools [214]. The privileges offered in the Russian Empire had a significant influence on the decision of many German speakers to leave their home country and settle in Russian territory, as they faced high taxes, military and frontline services, economic hardship, crop failures,

famines, infringement of religious freedom, and others challenges in their home country [215].

This initial situation implied a *conflict potential*, as the **inequality** between the now coexisting groups—the local population and the indigenous population—could be perceived as **unfair**. Furthermore, the Orthodox Church was hostile to the Germans' freedom of religion. **Linguistic and cultural differences** within the geographical area are also indicators of potential conflict.

Over the course of a century, the German-speaking groups settled in various regions of Russia and founded colonies, which were often named after their old cities, such as Darmstadt or Rosenberg [215]. There, and in the daughter colonies that emerged from them, they developed their own specific culture, language, and identity, which were heavily influenced by German traditions. Their identity and language were closely intertwined and played an important role in the community until the start of the Russification strategy under Emperor Alexander II. The Russification strategy of Alexander II, intensified mainly in the 1860s and 1870s, aimed to **assimilate** the non-Russian groups of the Russian Empire and suppress their cultural identity in favor of the Russian one through various measures. One such measure was language policy, which promoted Russian as the dominant language in educational institutions, administration, and public life, while other languages were increasingly suppressed or restricted. Consequently, in 1871, the perpetually guaranteed privileges of the Volga Germans were revoked.

This already reflects the *second and third stages of the E–D Pyramid*: Germans as an ethnic minority have their once guaranteed **privileges revoked** and are demanded to **adapt** to Russian life. To circumvent this conflict, many Germans had already **emigrated** to America, Canada, or South America by the outbreak of World War I [215]. Those Germans who remained in Russia had to adapt to the Russian system. The two indicators, **emigration** and **adaptation**, are particularly evident at this point as they serve as de-escalation strategies. They are to be classified at the highest stage of de-escalation, the *capitulation*.

Under Tsar Nicholas II in Russia, the **repressions** against the Volga Germans were intensified. This occurred within the framework of a general policy of suppression of non-Russian ethnic groups and minorities, which were considered potential threats to Russian unity. Some of the repressions experienced by the Volga Germans during the reign of Tsar Nicholas II included: prohibition of the use of the German language in public, banning of German newspapers and books, as well as **expulsions**. Additionally, in 1915, there were **expropriation laws**, **pogroms**, or forced measures such as **forced labor** against Germans in Moscow [216]. Already at this time, Germans were sporadically deported to remote areas of the Russian Empire, e.g. Siberia. These measures were often carried out arbitrarily and served to intimidate the Volga Germans and weaken their community. In 1917, these forced resettlements were temporarily halted due to the revolution. Nevertheless, a period of legal uncertainty followed as well as the famine, which also heavily affected the German settlers.

In summary, the situation for the Volga Germans worsened in the beginning of the 20[th] century. This time already reflects the *fourth stage of escalation in the E–D Pyramid*, the *violence*, as Germans were seen and degraded as "internal enemies" [217] and were therefore also subjected to **physical violence**. This *fourth stage of escalation* was reinforced by the assimilation policy of the Soviet government in the 1920s and 1930s.

The repressions against the Volga Germans reached a peak during World War II, which can also be seen as the *highest stage of escalation in the E–D Pyramid*. Under the leadership of Josef Stalin, the Volga Germans were viewed as potential collaborators of the enemy due to their ethnic background within the framework of the Soviet Union's general war efforts. The massive intensification of repressions included **deportations** and **forced labor**, **expropriation** and **expulsion**, **discrimination** and **abuse**, **forced recruitment**, as well as significant losses

within the Volga German population [218]. The entire German-speaking population from all the settled Russian territories was deported to Kazakhstan, Omsk, Novosibirsk, the Altai region, and neighbouring areas, so that by the end of the 1960s, only 8% of the Germans lived in the European part of the USSR [215]. This **genocide** of the Volga Germans was part of a broader policy of ethnic cleansing under Stalin's rule. Additionally, after the invasion of the Germans into the Soviet Union, resettlements to Germany occurred: approximately 20% of the Volga Germans were placed under the rule of the National Socialists under the slogan "Heim ins Reich" (Home to the Reich) [215]. Overall, the Volga Germans were severely oppressed during World War II and suffered from significant human rights violations.

After World War II, the Volga Germans continued to experience difficult times, but their situation slowly began to improve: **Diplomatic relations** between the Federal Republic of Germany and the Soviet Union were revived under Konrad Adenauer, and prisoners of war were liberated. Some of the Volga Germans were allowed to return from the deportation camps, although many of them had lost their homes and property. With the collapse of the Soviet Union and the opening of Russia in the 1990s, a phase of cultural revitalization began for the now predominantly Russian-speaking Volga Germans and their families. With the end of the Soviet Union and increasing freedom, the Volga Germans were able to cultivate their cultural identity more openly and actively. Societies, cultural organizations, and schools dedicated to promoting the German language and culture emerged. Many Volga Germans seized the opportunity to **emigrate** to their historical homeland with their Russian family members. This led to the formation of a significant Volga German diaspora in Germany and other Western countries. After the collapse of the Soviet Union, the Volga Germans also began to address their historical experience and the oppression during the Soviet era. Efforts were made to document the stories of the victims, establish **memorials**, and keep alive the memory of the suffering and hardship during the deportations and repressions.

The improvement of the situation of the Volga Germans during the post-war period represents individual stages of *de-escalation*. The main de-escalatory indicators in this case were the **diplomatic relations**, the **facilitated remigration** of the Volga Germans to their historical homeland, as well as the establishment of **memorials**.

Through the case study of the Volga Germans, it was illustrated how the individual stages of escalation can be applied to the German-speaking settlers on Russian territory. The escalation of the conflict was reflected in the various stages of the E–D Pyramid, from exclusion and assimilation to massive repression during World War II and the final genocide. Fig 17 shows the individual indicators. After World War II, a phase of de-escalation began, during which the Volga Germans slowly regained their rights and advocated for their historical recognition. The opening of Russia in the 1990s enabled a cultural revitalization and the formation of a significant diaspora in Germany and other countries.

## Results

As we approach the data extracted from the case studies, the first question that we would like to consider is whether there is any patterning in the time spans typically associated with the different levels of the pyramid. In order to answer this question we use the data underlying the ten illustrations of timelines in the previous section. Since the cases represent time scales that differ by an order of magnitude, from a 237 year total span in the Volga Germans case to a 2550 year span in the case of Sambia, we normalize the duration of each indicator by the total time span for the individual case and multiply the resulting value by 100 so that they can be thought of as percents of the total time spans. We subsequently pool time spans for all indicators pertaining to each pyramid cell across cases. An indicator is interpreted as pertaining to a

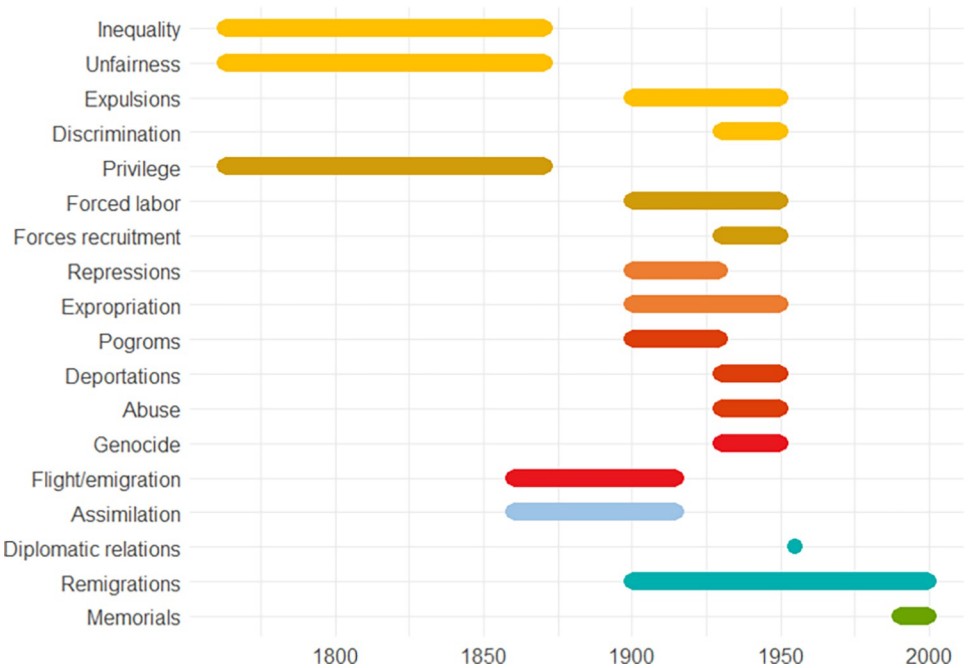

**Fig 17. Approximate chronology of the indicators described in the Volga Germans case study.**

cell whenever the confidence score assigned to it is greater than 0. That is, in this particular analysis we do not use confidence scores for any kind of differentiation. In this way, the 10 case studies yield between 21 and 112 normalized time spans for each cell in the pyramid (ignoring the absence of time spans for de-escalation level 5, which is never in play). The (rounded) means of these numbers are displayed in the pyramid cells to which they belong in Fig 18.

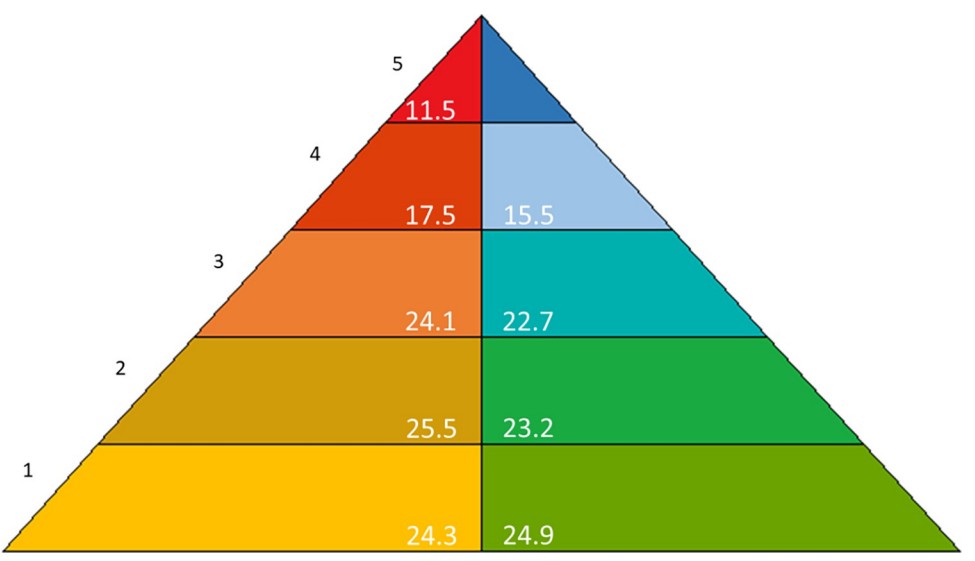

**Fig 18. Rounded values of mean normalized time spans in the different pyramid cells.**

Fig 18 shows that mean time spans get shorter towards the top of the pyramid, the only exception to this pattern being escalation level 1 (bottom left cell), whose value (24.3) should be greater than that of escalation level 2 (value: 25.5) but is smaller. Nevertheless, the nine values and their respective pyramid levels are solidly negatively correlated: $r = -0.89$. Thus, we can conclude that there is a tendency for indicators to have a shorter duration the higher the level of escalation or de-escalation is.

Looking at the values for identical levels of escalation and de-escalation in Fig 18, we see that the values of the escalation side are greater at the escalation side at all but the lowest level. None of the apparent differences in means are significant by a t-test, however. It is therefore more correct to say that there is no significant difference between the duration of escalation and de-escalation events pertaining to one and the same level.

Taken together, the results for durations of different indicators in the vertical and horizontal dimension provide a good proof of concept for the pyramid model: there is a significant gradient running from the bottom to the top of the pyramid, whereas cells that are claimed to be on the same level indeed behave similarly. The real-life implications of the differences in the vertical dimension will be an item for the Discussion section.

We now go in to compare the ten case studies. A way to summarize and visually compare them is by 'fingerprinting' indicator values (confidence scores) in the shape of Escalation–De-escalation Pyramids (E–D Pyramids). For this purpose a single value representing some summary statistic of the confidence scores of different indicators in each pyramid cell has to be chosen. Among different options that offer themselves we have considered the maximum, the sum, the mean, and the median. Choosing the maximum reduces the data to just one data point per pyramid cell, which seems unwise. The sum is a worthy alternative avoiding this kind of data reduction, but different cases become hard to compare when scales are different. The mean and the median have the advantage that they both operate within the scale of the original confidence scores, which runs from 0 to 1. In order to choose between them we have looked into the distribution of the confidence scores from different indicators within each pyramid cell. For the mean to be a meaningful summary statistic, the values should not stray too far from a normal distribution. We found, however, that by a Shapiro-Wilk test it was only one fifth of the distributions involving three or more non-identical values (i.e., cases satisfying the conditions for the test) that could be considered normal. Therefore, the median, which is less sensitive to skewed distributions, was adopted. For each case study and pyramid cell, then, we computed medians, and for the purposes of an initial appreciation, these medians are displayed as points on a white-to-black scale in Fig 19. Thus, rather than using colors symbolizing the different levels of escalation and de-escalation we now focus on the values of the (median) confidence scores within these different levels, indicating these values by shades of gray.

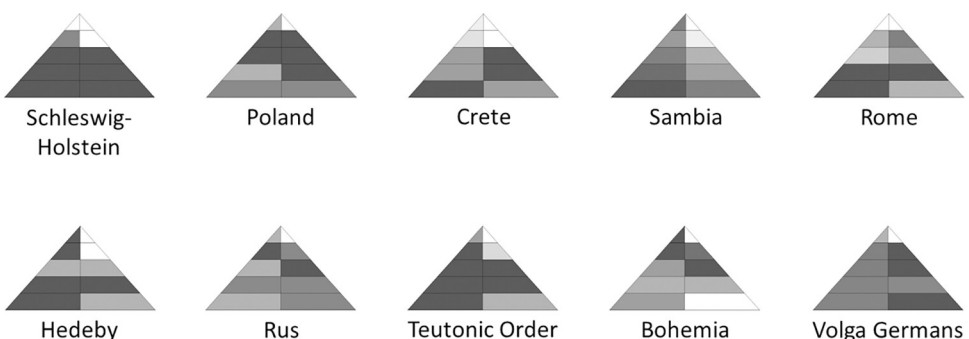

**Fig 19.** a-j. Conflict fingerprints based on medians for the different case studies.

When interpreting the fingerprints in Fig 19 it should be kept in mind that de-escalation level 5 (the top left of the pyramid) is never realized since there is (probably) no de-escalation counter-part to total annihilation. Thus, this cell should be ignored. Other white cells are also ignored, since zero values are equivalent to the absence of involvement of any indicator at the cell in question.

Visually the fingerprints in Fig 19 suggest quite a lot of diversity. This is good news inasmuch as we would hope to have captured a healthy portion of the possible variation in conflict situations, but it makes meaningful comparisons potentially challenging. In the following we present two comparisons, where one focuses on overall proportions of escalation to de-escalation, in a sense contrasting the two sides of the pyramid—the horizontal dimension—, and the other focuses on patterns between the levels—the vertical dimension.

By contrasting the two sides of the pyramid for a given case we can discern whether there is an overall balance or lack of balance in escalation and de-escalation during the time span represented by the case. We do this by weighting the values in the cells on each side of the pyramid by numbers running from 1 to 5, corresponding to the levels from bottom to top of the pyramids. Subsequently we average the values belonging to either side and plot them against each other, cf. Fig 20. The red line in the plot represents the diagonal. The closer to the diagonal a case appears, the greater the balance between escalative and de-escalative indicators. Several cases show a high degree of balance (Rus, Poland, Bohemia), while some have a high degree of skewing towards escalation (Sambia, Hedeby, Teutonic Order) and others more of a skewing towards de-escalation (Rome, Crete). Yet others are slightly unbalanced in one or the other direction (Schleswig-Holstein, Volga Germans).

The weighted means approach that was used for the analysis is simple and convenient. We are aware that this method also has disadvantages since the distribution of confidence scores along the five levels can be quite different and still yield the same weighted means. The advantage of the simplicity of this method should largely outweigh these disadvantages.

Going on to look more at the vertical dimension, we will observe whether there is a gradiant in the escalation or de-escalation half and in what direction it runs. Schleswig-Holstein and

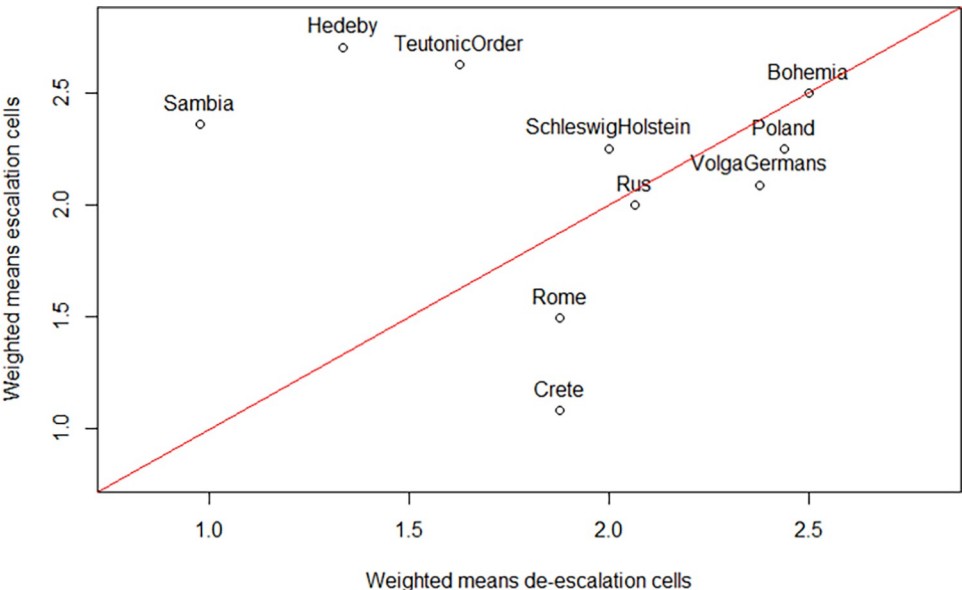

**Fig 20. Weighted means of escalation and de-escalation cells plotted against each other.**

Sambia generally show greater intensity towards the bottom in both halves of the pyramid, while for Bohemia the picture is more of the opposite, generally with greater intensity towards the top in both halves. For Crete there is a gradient in the left half (escalation) from greater to smaller confidence scores moving from bottom to top, and the right half tends towards a mirror image. Interestingly, there is no good candidate for the opposite of the Crete picture where the intensity would increase towards the top for escalation and decrease towards the bottom for de-escalation. Nevertheless, we can carve out four 'ideal types' (to borrow Max Weber's term) that are characterized by values of either 1 or -1 for the two following correlations: between pyramid levels and escalation confidence scores and between pyramid levels and de-escalation confidence scores. In the real world we might not expect any of the ideal types to be realized, but the types represent a field of possibilities convenient for describing and interpreting our findings.

The four corners of the plot in Fig 21 represents the four ideal types just characterized. The plot shows Spearman's Rank Correlation Coefficient between escalation levels (from 1 to 5

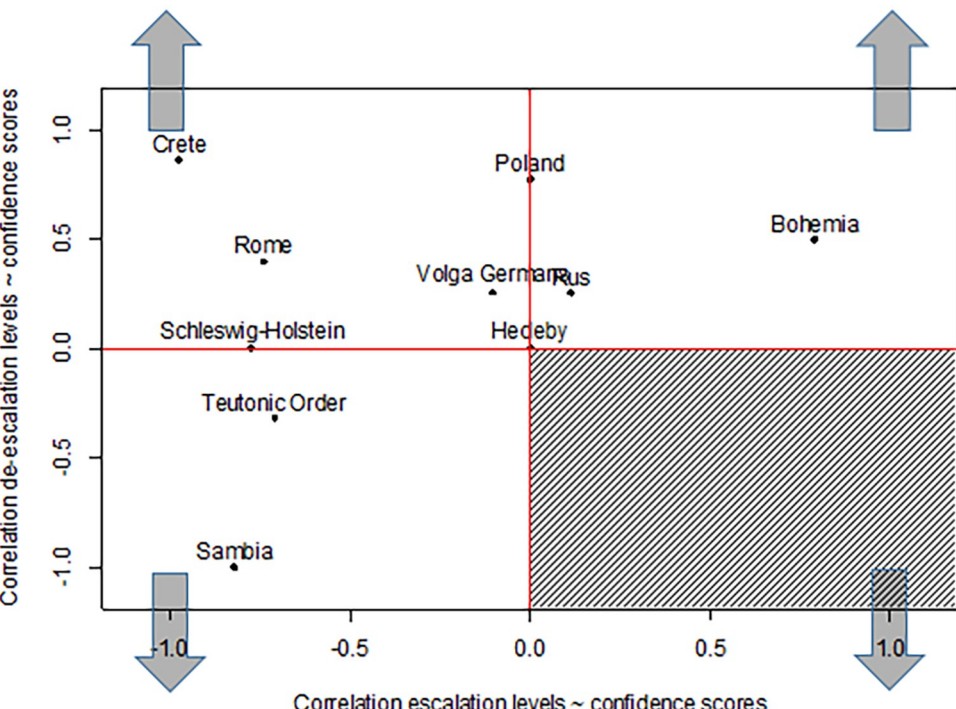

**Fig 21. Spearman correlations between median confidence scores in pyramid cells and escalation and de-escalation levels, respectively.** The red lines separate the four major regions of the plot.

going from bottom to top) and confidence scores on the horizontal axis and the corresponding correlations between de-escalation levels and confidence scores on the vertical axis. The absence of cases pertaining to the lower right quadrangle (indicated by hatching) potentially has important implications, which we take up in the Discussion section.

## Discussion

The case studies representing the empirical basis of this study illustrate that many aspects of the research were left up to the intutitions of individual researchers. One such aspect is how to delimit a given conflict situation in the first place. We do not offer guidelines for this. For instance, we do not try to define the sense in which Bronze Age Crete may be regarded as *a* conflict situation. Nor do we attempt to heavily restrict the assignment of different indicators to different pyramid cells. It is possible that a greater rigidity of definitions and procedures could be an asset in the future, but in a first exploration it seems advantageous to take a generous approach. Moreover, we found in the Results section that there is enough systematicity in the data that meaningful patterns emerge. Thus, in spite of the fact that pyramid levels are not very strictly defined, we found that the duration of indicators is negatively correlated with the pyramid level whereas the duration is similar for escalation and de-escalation when the level is the same. Working our way up from the bottom of the pyramid we can easily account for this pattern applying common sense (we will use e1-e5 to refer to escalation levels and d1-d4 to refer to de-escalation levels). Conflict potential (e1) and cooperation (d1) are basic ingredients in social interaction that are omnipresent and expected to often be long-lived. Interfering demands (e2) and regulation (d2) are more subject to fluctuations, but it comes as no surprise that they may also be long-lived. Threats (e3) and conciliation/negotiation (d3) inherently have shorter lives, as they will tend to give way to other levels. For instance, threats may escalate into violence (e4) and negotiation may lead to regulations (d2). Physical violence (e4) is inherently short-lived, probably because it will relatively quickly lead to a reaction, implying a change of levels, and the same would be true of capitulation. Extinction (e5) has to be relatively quick to be efficiently accomplished.

In addition to providing a proof of concept for the E–D Pyramid model, the result that the duration of a situation tends to decrease as the intensity increases is interesting in its own right. While it is nice to see a quantitative confirmation we are not surprised that outbreaks of violence come out as short-lived in comparison to the omnipresent mere potential of conflict, but the insight that there are intermediate levels of intensity of conflicts both at the escalation and de-escalation sides which also show a relationship between time and intensity, is, as far as we are aware, completely new.

The lack of rigidity in the application of the E–D Pyramid model apparently also did not prevent patterns from emerging when comparing different cases. As one result we observed that different conflicts can be compared in terms of their degree of balance between escalation and de-escalation. We see this result as providing a more sophisticated and therefore better alternative to a characterization of situations in terms of peaceful vs. conflictive or warlike. The latter kind of characterization is mono-dimensional whereas ours is two-dimensional. In a mono-dimensional description our cases might be ordered on a single line according to the amount of violence displayed, which is clearly a reductionist metric.

The kind of comparison that we offered in Fig 20 might inspire further inquiries into possible consequences of different degrees of balance for other historical developments. For instance, it is hard not to begin to ponder the fact that Rome and Crete had a great impact on European culture history and were also associated with an imbalance towards de-escalation measures.

**Table 4. Truth table for p → q and distribution of combinations of clines of confidence scores in the vertical dimension of the E–D Pyramid.**

| P | Q | p → q | de-escalation towards top | escalation towards top | combination present |
|---|---|---|---|---|---|
| T | T | T | decreasing | decreasing | ✓ |
| T | F | F | decreasing | increasing | ÷ |
| F | T | T | increasing | deecreasing | ✓ |
| F | F | T | increasing | increasing | ✓ |

We also found some meaningful results for the comparison of clines along the vertical dimensions of E–D pyramids, cf. Fig 21. While subtle, this may actually be the most important empirical result for conflict research of the present paper. Stated plainly, we found that the confidence scores may increase or decrease toward the top of the pyramid with escalation and de-escalation behaving in the same way, or the confidence scores for escalation may decrease towards the top of the pyramid while the confidence scores for de-escalation increase. What we do not see are cases where the confidence scores for de-escalation decrease towards the top of the pyramid while the confidence scores for escalation increase in the same direction. Thus, there is a material implication here, as illustrated in Table 4. The table shows the well-known truth table for a material implication p → q compared with the equivalent distribution of presences and absences of combinations of directions of clines of escalation and de-escalation. By the logic of the material implication, de-escalation having greater confidence scores towards the bottom of the E–D Pyramid implies that escalation increases towards the bottom of the pyramid. The implication is a hypothesis to be more firmly established with the addition of more case studies. We expect that counterexamples may be found but that the implication may still appear as a strong tendency in a statistical sense. If so, this would suggest that if one wants to avoid escalation with greater involvement of the higher levels, including physical violence, it is best to develop a situation where the lower levels of de-escalation, including cooperation and regulation are firmly in place.

We do want to add, though, that these results stem from a model which represents a strong simplification of reality inasmuch as the diachronic dimension has been collapsed and causal relationships, although broadly assumed to exist, have not been established or explored. Further research applying a dynamized version of the E–D Pyramid model as well as more data would be necessary for confirmation of the findings.

## Conclusions

This paper has offered both theoretical-methodological and empirical approaches to conflict research from the perspective of the study of the past (archaeology and history). We introduced a model embracing both escalation and de-escalation aspects of a conflict situation designed to yield simplified descriptions ('fingerprints' in our terminology) of a particular situation that allow for quantifiable comparisons with other situations. The application of the model, which we refer to as the Escalation–De-escalation Pyramid, was thoroughly exemplified through ten relatively detailed case studies sampled from the European past between the Bronze Age and the the 20[th] century. In addition to the narratives pertaining to each case, the experts in our team supplied lists of conflict factors as manifested by different kinds of evidence—'indicators' in our terminology. The indicators were all assigned confidence scores indicating the amount of confidence in their interpretation, which is also an expression of their intensity or saliency. Using this data we were able to present analyses suggesting that (1) indicators tend to have a shorter duration the closer they are to the highest levels of escalation or de-escalation; (2) different situations can meaningfully and usefully be characterized in

terms of the degree of balance between escalation and de-escalation; (3) different combinations of gradients in escalation or de-escalation exist, but we do not observe a combination of strongly represented lower escalation levels and strongly represented higher levels of escalation, suggesting the hypothesis that the safest way to avoid high levels of escalation like outbreaks of violence may be to focus on basic (lower) levels of de-escalation like cooperation and regulation.

We hope that the promising results of these quantitative analyses may inspire further research using the E–D Pyramid model and, in general, will stimulate further research of a comparative nature into past and present conflicts.

## Acknowledgments

The research presented here pertains to the Subcluster "Conflict: Competition and Conciliation" within the ROOTS Cluster of Excellence at Kiel University. We would like to acknowledge contributions to discussions leading to the paper by the remaining members of the Subcluster.

## Author Contributions

**Conceptualization:** Søren Wichmann, Anna K. Loy, Anna-Theres Andersen, Ralf Bleile, Darja Jonjić, Jutta Kneisel, Norbert Nübler, Andrea Santamaria, Jens Schneeweiß, Gerald Schwedler, Katharina Zerzeropulos, Lorenz Kienle, Oliver Nakoinz.

**Formal analysis:** Søren Wichmann, Oliver Nakoinz.

**Funding acquisition:** Lorenz Kienle.

**Investigation:** Søren Wichmann, Anna K. Loy, Anna-Theres Andersen, Ralf Bleile, Darja Jonjić, Jutta Kneisel, Norbert Nübler, Andrea Santamaria, Jens Schneeweiß, Gerald Schwedler, Katharina Zerzeropulos, Oliver Nakoinz.

**Methodology:** Søren Wichmann, Anna K. Loy, Oliver Nakoinz.

**Project administration:** Lorenz Kienle.

**Software:** Søren Wichmann, Andrea Santamaria, Oliver Nakoinz.

**Visualization:** Søren Wichmann, Jutta Kneisel, Andrea Santamaria, Oliver Nakoinz.

**Writing – original draft:** Søren Wichmann, Anna K. Loy, Anna-Theres Andersen, Ralf Bleile, Darja Jonjić, Jutta Kneisel, Norbert Nübler, Andrea Santamaria, Jens Schneeweiß, Gerald Schwedler, Katharina Zerzeropulos, Oliver Nakoinz.

**Writing – review & editing:** Søren Wichmann, Lorenz Kienle.

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
