## [Decision Letter · Decision Letter 0]

16 Sep 2024

PONE-D-24-21063Fingerprinting conflict: A comparative model with applications to archaeological and historical dataPLOS ONE

Dear Dr. Wichmann,

Thank you for submitting your manuscript to PLOS ONE. After careful consideration, we feel that it has merit but does not fully meet PLOS ONE’s publication criteria as it currently stands. Therefore, we invite you to submit a revised version of the manuscript that addresses the points raised during the review process.

The attached reviews offer several important observations and questions. In light of this feedback, revisions to the manuscript are necessary. Please carefully address all comments, particularly those from Reviewer 1. Given the length of the paper and the extensive literature on violence, I recommend refraining from adding new references to your bibliography. However, please ensure full compliance with all other comments from Reviewer 2.

We look forward to receiving your revised manuscript.

Kind regards,

Stefanos Gimatzidis, Ph.D.

Academic Editor

PLOS ONE

“This work was supported by the Deutsche Forschungsgemeinschaft (German Research Foundation) under Germany’s Excellence Strategy (grant EXC 2150 390870439).”

3. We note that Figures 9 and 10 in your submission contain copyrighted images. All PLOS content is published under the Creative Commons Attribution License (CC BY 4.0), which means that the manuscript, images, and Supporting Information files will be freely available online, and any third party is permitted to access, download, copy, distribute, and use these materials in any way, even commercially, with proper attribution. For more information, see our copyright guidelines: http://journals.plos.org/plosone/s/licenses-and-copyright.

1. You may seek permission from the original copyright holder of Figures 9 and 10 to publish the content specifically under the CC BY 4.0 license.

Reviewers' comments:

Reviewer's Responses to Questions

**Comments to the Author**

1. Is the manuscript technically sound, and do the data support the conclusions?

Reviewer #1: Yes

Reviewer #2: Partly

2. Has the statistical analysis been performed appropriately and rigorously? 

Reviewer #1: I Don't Know

Reviewer #2: Yes

3. Have the authors made all data underlying the findings in their manuscript fully available?

Reviewer #1: Yes

Reviewer #2: Yes

4. Is the manuscript presented in an intelligible fashion and written in standard English?

Reviewer #1: Yes

Reviewer #2: Yes

5. Review Comments to the Author

Reviewer #1: Review

The article consists of several parts that build on each other in a comprehensible way. The first part explains why a new model was developed to describe conflicts in the past. The authors rightly criticize the fact that the discussion has increasingly focused on war and violence in the last years. At least from the perspective of archaeology, I can confirm this. They propose a model in which de-escalation and escalation are examined in order to comparatively investigate (pre-)historical conflict processes. Indicators for de-escalation and escalation are presented as examples and categorized into five levels of a pyramid.

The second part includes ten case studies. They are described on the basis of indicators of de-escalation and escalation, and the archaeological and historical sources are presented. As the archaeological and historical evidence of the individual case studies is very different, the indicators are also heterogeneous. The authors are aware of this and characterize their approach as "generous": "Nor do we attempt to heavily restrict the assignment of different indicators to different pyramid cells" (lines 1928-29). For each case study, the indicators are listed along an approximate timeline and each indicator is assigned to a specific level of the pyramid. The case studies are clearly presented. Only the case study 'trade policies' in Rome stands out. Here the focus is exclusively on the economic sphere and the economic policies are described in lines 1207-1208 as a potential for conflict. It is thus limited to only one aspect of Roman society in the period of the Early Republic and Late Empire, whereas all other case studies attempt to take into account sources from various areas of (pre-)historical communities and processes.

The section with the case studies is purely descriptive but provides an interesting approach to how completely different times and regions can be described using the model of de-escalation and escalation. But is this model also suitable as an analytical tool? The authors address this question in the third and final part of the article. They evaluate the indicators in combination with the levels in the pyramid using several statistical methods. The result of the summary statistics is the 'fingerprinting' they also refer to in the title. It is shown in Figure 21 and is easy to understand except for the sometimes difficult to distinguish gray shades.

In connection with 'fingerprinting', the article takes a temporal factor into account, but only to make the levels in the pyramid statistically comparable. The temporal dimension is my main criticism of this manuscript, as it is not discussed critically enough. The individual case studies show various processes or events that are classified as de-escalating or escalating. However, in most cases it is not clear whether and to what extent de-escalating and escalating indicators are connected in time or even represent a reaction within particular processes. In the case study of the Volga-Germans, for example, indicators of escalation can be found up to 1875, while there are no indicators of de-escalation in the same period. Does this mean that no de-escalation measures were taken until well into the 19th century? Generally, the question arises how and to which extent certain indicators are connected in time and may have led to an increase/decrease in escalation and de-escalation. Direct correlations can certainly only ever be assumed - especially in prehistoric times. But I think this is an important point that the authors should discuss critically because such considerations are particularly interesting for the interpretation of the following statistical analyses. Fig. 22 shows the result of the analysis of the weighted means, but here too the indicators for de-escalation and escalation were evaluated in the respective case studies, always taking the entire period into account.

A revision of Figure 23 ‘Spearman correlation’ is advisable, as 'Schleswig-Holstein' is missing. Secondly, the four 'ideal types' in the same figure should be added graphically to make it easier to understand. The following conclusion is derived from this analysis: "different combinations of gradients in escalation or de-escalation exist, but we do not observe a combination of strongly represented lower escalation levels and strongly represented higher levels of escalation, suggesting the hypothesis that the safest way to avoid high levels of escalation like outbreaks of violence may be to focus on basic (lower) levels of de-escalation like cooperation and regulation" (lines 1996-1999). I consider this to be problematic, as the temporal and causal relationships for the case studies have not been discussed in detail or are often not clear from the archaeological and historical sources.

This is of course a problem that is inherent to the sources and cannot be solved. But I would therefore advise the authors to comment in detail on the temporality of the indicators. The aim stated in the abstract is 'to identify typical patterns in conflict situations'. They are still formulated very abstractly in this manuscript, however convincing the 'finger printing' in the case studies may be.

Some suggestions for corrections:

Keywords: delete ‘war’ and ‘peace’, replace with ‚escalation‘ and ‚de-escalation’

line 49: ‘Finally we offer a Conclusion’, please contribute a little bit more to this sentence.

line 238: ‘but it also be imply …’ Delete ‘be’

line 268: two times ‘representing’

Lanzenspitzen, better ‘spearheads’ (please check)

line 426: ‘burial constructions’, not ‘custom’ when you are writing about burial mounds

Case study: Poland: in the text you discuss cooking pits as gathering places, but you do not show them as indicators of de-escalation in Fig. 7?

line 604: ‘the subsistence strategy changed’ instead of ‘a changed subsistence strategy emerged’

Case study: Hedeby: Please use the same terms in the text and in Fig. 15. In Fig. 15 two times ‘signal fires’. The first time it is given as an indicator of escalation, but without a symbol

Case study: Rus’: Fig. 16 – Why are the events of ‘Byzantine military services’ not given in one line according to the time line?

line 1482-38 and Fig. 16: familiar missionary language: should it be assigned only descalating, according to the text?

Reviewer #2: I provided detailed comments to the paper in a word file I uploaded. The review opens with a general comment on the paper. My primary concern is the simplicity of the model since the studied social phenomenon is more complex. I then provide an example demonstrating my point and continue with detailed comments to improve the quality of the paper.

6. PLOS authors have the option to publish the peer review history of their article (what does this mean?). If published, this will include your full peer review and any attached files.

Reviewer #1: No

Reviewer #2: No

---

## [Author Response · Author response to Decision Letter 0]

18 Oct 2024

Below we indicate (in the paragraphs followed by ***) how we have revised the paper in response to suggestions from the editorial office of the journal and the two reviewers (unmarked). This text is also to be found in the document called Response to Reviewers.

Comments from the editorial office

***Done.

“This work was supported by the Deutsche Forschungsgemeinschaft (German Research Foundation) under Germany’s Excellence Strategy (grant EXC 2150 390870439).”

***Done.

***Done.

3. We note that Figures 9 and 10 in your submission contain copyrighted images. All PLOS content is published under the Creative Commons Attribution License (CC BY 4.0), which means that the manuscript, images, and Supporting Information files will be freely available online, and any third party is permitted to access, download, copy, distribute, and use these materials in any way, even commercially, with proper attribution. For more information, see our copyright guidelines: http://journals.plos.org/plosone/s/licenses-and-copyright.

***We have removed the figures, as per a suggestion from Reviewer #2. We agree with the Reviewer that they are not essential.

***The reference list has been reviewed. No retracted papers are cited.

Comments from reviewers

Reviewer #1: Review

The article consists of several parts that build on each other in a comprehensible way. The first part explains why a new model was developed to describe conflicts in the past. The authors rightly criticize the fact that the discussion has increasingly focused on war and violence in the last years. At least from the perspective of archaeology, I can confirm this. They propose a model in which de-escalation and escalation are examined in order to comparatively investigate (pre-)historical conflict processes. Indicators for de-escalation and escalation are presented as examples and categorized into five levels of a pyramid.

The second part includes ten case studies. They are described on the basis of indicators of de-escalation and escalation, and the archaeological and historical sources are presented. As the archaeological and historical evidence of the individual case studies is very different, the indicators are also heterogeneous. The authors are aware of this and characterize their approach as “generous”: “Nor do we attempt to heavily restrict the assignment of different indicators to different pyramid cells” (lines 1928-29). For each case study, the indicators are listed along an approximate timeline and each indicator is assigned to a specific level of the pyramid. 

***Thank you for an accurate characterization of our paper.

The case studies are clearly presented. Only the case study ‘trade policies’ in Rome stands out. Here the focus is exclusively on the economic sphere and the economic policies are described in lines 1207-1208 as a potential for conflict. It is thus limited to only one aspect of Roman society in the period of the Early Republic and Late Empire, whereas all other case studies attempt to take into account sources from various areas of (pre-)historical communities and processes.

***In the beginning of the text for this case study we added a bit of text stressing that we are being selective, not trying to be comprehensive.

The section with the case studies is purely descriptive but provides an interesting approach to how completely different times and regions can be described using the model of de-escalation and escalation. But is this model also suitable as an analytical tool? The authors address this question in the third and final part of the article. They evaluate the indicators in combination with the levels in the pyramid using several statistical methods. The result of the summary statistics is the ‘fingerprinting’ they also refer to in the title. It is shown in Figure 21 and is easy to understand except for the sometimes difficult to distinguish gray shades.

***Again thank you for an accurate characterization. It is not important to distinguish the shades in detail. The underlying data are available if precise numbers are needed.

In connection with 'fingerprinting', the article takes a temporal factor into account, but only to make the levels in the pyramid statistically comparable. The temporal dimension is my main criticism of this manuscript, as it is not discussed critically enough. The individual case studies show various processes or events that are classified as de-escalating or escalating. However, in most cases it is not clear whether and to what extent de-escalating and escalating indicators are connected in time or even represent a reaction within particular processes. In the case study of the Volga-Germans, for example, indicators of escalation can be found up to 1875, while there are no indicators of de-escalation in the same period. Does this mean that no de-escalation measures were taken until well into the 19th century? Generally, the question arises how and to which extent certain indicators are connected in time and may have led to an increase/decrease in escalation and de-escalation. Direct correlations can certainly only ever be assumed - especially in prehistoric times. 

***Establishing causation in history is a can of worms. But actually we currently keep a distance to this problem, although this was perhaps not made clear enough. Now we added the following text to the end of the section called ‘The model’: “While it is fair to assume that for a given conflict there is some degree of interrelatedness of actions corresponding to the various indicators observed, i.e. a network of causality—perhaps with some events that are more central than others—it not a requirement of the model that two indicators are in fact related. For instance, in a given conflict some act of threat (escalation Level 3) may be identified as well as some act of violence (Level 4), and both would then be registered in the descriptive model. But the description is a static one, so there is no necessity of an assumption of direct causation between the act of threat and the act of violence. In a further development of the model a dynamic perspective may conceivably be introduced, allowing for the identification of diachronic patterns of conflict developments. This is a matter for future work, however.”

But I think this is an important point that the authors should discuss critically because such considerations are particularly interesting for the interpretation of the following statistical analyses. Fig. 22 shows the result of the analysis of the weighted means, but here too the indicators for de-escalation and escalation were evaluated in the respective case studies, always taking the entire period into account.

***An analysis such as the one in Fig. 22 is consistent with our static approach, which is now made more explicit in the text added in response to the previous comment. We are aware that the approach is reductionist inasmuch as the diachronic perspective is collapsed. Since we already state once that a dynamization of the model is (potentially) a matter of future research we won’t repeat that in the text accompanying Fig. 22, but we are aware of the limitations.

A revision of Figure 23 ‘Spearman correlation’ is advisable, as ‘Schleswig-Holstein’ is missing. 

***Schleswig-Holstein was missing because of an absence of a correlation in the de-escalation dimension, which was interpreted by the R function as NA (“not applicable”). This was replaced by a zero, now allowing for the inclusion of Schleswig-Holstein. Thank you for this very acute observation.

Secondly, the four ‘ideal types’ in the same figure should be added graphically to make it easier to understand. 

***We have now amended the figure with annotations attached to the four corners and more verbose axis labels to make it easier to understand.

The following conclusion is derived from this analysis: “different combinations of gradients in escalation or de-escalation exist, but we do not observe a combination of strongly represented lower escalation levels and strongly represented higher levels of escalation, suggesting the hypothesis that the safest way to avoid high levels of escalation like outbreaks of violence may be to focus on basic (lower) levels of de-escalation like cooperation and regulation” (lines 1996-1999). I consider this to be problematic, as the temporal and causal relationships for the case studies have not been discussed in detail or are often not clear from the archaeological and historical sources.

This is of course a problem that is inherent to the sources and cannot be solved. But I would therefore advise the authors to comment in detail on the temporality of the indicators. The aim stated in the abstract is 'to identify typical patterns in conflict situations'. They are still formulated very abstractly in this manuscript, however convincing the 'finger printing' in the case studies may be.

***We now appended the following remarks to the end of the Discussion section, again raising issues also discussed in new text at the end of the section ‘The model’.

“We do want to add, though, that these results stem from a model which represents a strong simplification of reality inasmuch as the diachronic dimension has been collapsed and causal relationships, although broadly assumed to exist, have not been established or explored. Further research applying a dynamized version of the E–D Pyramid model as well as more data would be necessary for confirmation of the findings.”

Some suggestions for corrections:

Keywords: delete ‘war’ and ‘peace’, replace with ‚escalation‘ and ‚de-escalation’

***Done.

line 49: ‘Finally we offer a Conclusion’, please contribute a little bit more to this sentence.

***This and surrounding text was deleted entirely, following the recommendation of Reviewer #2.

line 238: ‘but it also be imply …’ Delete ‘be’

***Done.

line 268: two times ‘representing’

***The second occurrence now replaced by “implying”.

Lanzenspitzen, better ‘spearheads’ (please check)

***We couldn’t find the word ‘Lanzenspitzen’ in our text, but in line 406 we used ‘lance points’. This has been changed to ‘lance heads’. 

line 426: ‘burial constructions’, not ‘custom’ when you are writing about burial mounds

***Done.

Case study: Poland: in the text you discuss cooking pits as gathering places, but you do not show them as indicators of de-escalation in Fig. 7?

***As was/is mentioned in the text (line 533 of the original submission) there is a lack of studies of the phenomenon for Poland, so we hesitated to make cooking pits an element in the formal analysis. We also now downplayed them by removing ‘cooking pits’ from the title of the case study.

line 604: ‘the subsistence strategy changed’ instead of ‘a changed subsistence strategy emerged’

***Done.

Case study: Hedeby: Please use the same terms in the text and in Fig. 15. In Fig. 15 two times ‘signal fires’. The first time it is given as an indicator of escalation, but without a symbol

***The list of indicators in the text and the designations as they appear in Fig. 15 are now the same and ‘signal fires’ appears only once. A lack of symbols (=timeline) occurs when a given indicator cannot be dated. This was/is mentioned in the caption to the figure.

Case study: Rus’: Fig. 16 – Why are the events of ‘Byzantine military services’ not given in one line according to the time line?

***This and some other indicators that appear discontinuously in the record are put on separate lines partly for technical reasons, partly because circumstances might be different.

line 1482-38 and Fig. 16: familiar missionary language: should it be assigned only descalating, according to the text?

***Fig. 16 shows a grey color, reflecting the fact that both de-escalation level 2 and escalation level 2 received the highest confidence (a ‘1’). As correctly observed, the thinking behind this scoring is not reflected in the text. To the statement “The linguistic proximity between early East Slavic and Old Church Slavonic would have had a positive effect on the willingness to accept Christianity” we therefore now add “but language standardization usually also comes with a cost of disputes over linguistic norms.”

Thank you for some very insightful comments!

Reviewer #2

The paper Fingerprinting conflict is a valuable study which should be published after major revisions. The following comments are meant to enhance the quality of the paper and inspire the authors to consider greater complexity of the social phenomenon they studied. Surely conflicts cannot be lumped into side 1 vs. side 2, since there are more often than not other involved interest groups, sometimes directly, sometimes indirectly involved. Within each supposed sides there are individuals and groups with different and conflicting interests during the conflict, both in escalation and de-escalation phases. Historical examples are numerous. 

For example, Second Intermediate Period Egypt (ca. 1650-1550 BCE) in which we have at least three conflicted sides, Thebans, Hyksos and Kushites. The Kushites in one phase of the conflict even involved other African polities as their allies (for example Medjay peoples of Eastern Desert and the Land of Punt from the Horn of Africa). Each side consisted of individuals and groups with different interests. For example, some Egyptians living in Lower Nubian military forts, that previously belonged to Egypt, simply changed sides and became loyal to the king of Kush. Would this be an example of de-escalation, changing sides? At the same time Hyksos offer the Kushites to join them in crushing the Thebans. Is this escalation or de-escalation and for whom? For Kushites this would be escalation since they would be dragged into a conflict of others. For Hyksos this would be de-escalation through victory over the Thebans. 

***This is really interesting. We would have loved to include a Egyptian case study if your expertise had been available to us. Thank you for bringing a concrete example to the table. We completely agree that modelling should take into account the possibility of multiple partners as well as allow for multiple perspectives. In a paragraph added a couple of pages from the end of the section called ‘The model’ we now discuss these issues as follows (reflecting your example but in more abstract terms since we lack expertise to properly present the concrete case that you mention): “… there is nothing that hinders an application of the model to more complex scenarios involving multiple partners. Interpretations then also become more complicated and perhaps more prone to 

---

## [Editor Report · Decision Letter 1]

4 Nov 2024

Fingerprinting conflict: A comparative model with applications to archaeological and historical data

PONE-D-24-21063R1

Dear Dr. Wichmann,

We’re pleased to inform you that your manuscript has been judged scientifically suitable for publication and will be formally accepted for publication once it meets all outstanding technical requirements.

Kind regards,

Stefanos Gimatzidis, Ph.D.

Academic Editor

PLOS ONE
---

## [Editor Report · Acceptance letter]

11 Nov 2024

PONE-D-24-21063R1 

PLOS ONE

Dear Dr. Wichmann, 

I'm pleased to inform you that your manuscript has been deemed suitable for publication in PLOS ONE. Congratulations! Your manuscript is now being handed over to our production team.

Kind regards, 

on behalf of

Dr. Stefanos Gimatzidis 

Academic Editor

PLOS ONE